# A spontaneous complex structural variant in *rcan-1* increases exploratory behavior and laboratory fitness of *Caenorhabditis elegans*

Yuehui Zhao[1], Lijiang Long[1,2], Jason Wan[3], Shweta Biliya[1], Shannon C. Brady[4], Daehan Lee[4], Akinade Ojemakinde[1], Erik C. Andersen[4], Fredrik O. Vannberg[1,5], Hang Lu[5,6], Patrick T. McGrath[1,2,6,7] *

1 School of Biological Sciences, Georgia Institute of Technology, Atlanta, Georgia, United States of America, 2 Interdisciplinary Graduate Program in Quantitative Biosciences, Georgia Institute of Technology, Atlanta, Georgia, United States of America, 3 The Wallace H. Coulter Department of Biomedical Engineering, Georgia Institute of Technology and Emory University, Atlanta, Georgia, United States of America, 4 Department of Molecular Biosciences, Northwestern University, Evanston, Illinois, United States of America, 5 Parker H. Petit Institute of Bioengineering and Bioscience, Georgia Institute of Technology, Atlanta, Georgia, United States of America, 6 School of Chemical & Biomolecular Engineering, Georgia Institute of Technology, Atlanta, Georgia, United States of America, 7 School of Physics, Georgia Institute of Technology, Atlanta, Georgia, United States of America

* patrick.mcgrath@biology.gatech.edu

**Data Availability Statement:** All RNA-seq and resequencing files are available from the SRA database NIH BioProject PRJNA526525.

## Abstract

Over long evolutionary timescales, major changes to the copy number, function, and genomic organization of genes occur, however, our understanding of the individual mutational events responsible for these changes is lacking. In this report, we study the genetic basis of adaptation of two strains of *C. elegans* to laboratory food sources using competition experiments on a panel of 89 recombinant inbred lines (RIL). Unexpectedly, we identified a single RIL with higher relative fitness than either of the parental strains. This strain also displayed a novel behavioral phenotype, resulting in higher propensity to explore bacterial lawns. Using bulk-segregant analysis and short-read resequencing of this RIL, we mapped the change in exploration behavior to a spontaneous, complex rearrangement of the *rcan-1* gene that occurred during construction of the RIL panel. We resolved this rearrangement into five unique tandem inversion/duplications using Oxford Nanopore long-read sequencing. *rcan-1* encodes an ortholog to human RCAN1/DSCR1 calcipressin gene, which has been implicated as a causal gene for Down syndrome. The genomic rearrangement in *rcan-1* creates two complete and two truncated versions of the *rcan-1* coding region, with a variety of modified 5' and 3' non-coding regions. While most copy-number variations (CNVs) are thought to act by increasing expression of duplicated genes, these changes to *rcan-1* ultimately result in the reduction of its whole-body expression due to changes in the upstream regions. By backcrossing this rearrangement into a common genetic background to create a near isogenic line (NIL), we demonstrate that both the competitive advantage and exploration behavioral changes are linked to this complex genetic variant. This NIL strain does not phenocopy a strain containing an *rcan-1* loss-of-function allele, which suggests that the residual expression of *rcan-1* is necessary for its fitness effects. Our results demonstrate how colonization of new environments, such as those encountered in the laboratory, can create

**Funding:** This work was supported by NIH GM114170 (to P.T.M), a John N. Nicholson fellowship (to S.C.B), an NSF CAREER Award (to E. C.A.), and NIH NS096581, GM088333, AG056436 (to H.L.). The John Nicholson Fellowship URL is: https://www.tgs.northwestern.edu/funding/fellowships-and-grants/internal-fellowships/nicholson-fellowship.html. The funders had no role in study design, data collection and analysis, decision to publish, or preparation of the manuscript.

**Competing interests:** The authors have declared that no competing interests exist.

evolutionary pressure to modify gene function. This evolutionary mismatch can be resolved by an unexpectedly complex genetic change that simultaneously duplicates and diversifies a gene into two uniquely regulated genes. Our work shows how complex rearrangements can act to modify gene expression in ways besides increased gene dosage.

## Author summary

Evolution acts on genetic variants that modify phenotypes that increase the likelihood of staying alive and passing on these genetic changes to subsequent generations (i.e. fitness). There is general interest in understanding the types of genetic variants that can increase fitness in specific environments. One route that fitness can be increased is through changes in behavior, such as finding new food sources. Here, we identify a spontaneous genetic change that increases exploration behavior and fitness of animals in laboratory environments. Interestingly, this genetic change is not a simple genetic change that deletes or changes the sequence of a protein product, but rather a complex structural variant that simultaneously duplicates the *rcan-1* gene and also modifies its expression in a number of tissues. Our work demonstrates how a complex structural change can duplicate a gene, modify the DNA control regions that determine its cellular sites of action, and confer a fitness advantage that could lead to its spread in a population.

## Introduction

Structural variation, resulting in the removal, duplication, insertion, or rearrangement of large ($>$ 50bp) genomic regions, makes up a significant component of natural genetic variation in many different species [1–6]. The largescale rearrangement of DNA can truncate genes, modify transcriptional regulatory regions, and/or increase gene dosage and expression. Consequently, structural variation can have profound, detrimental effects on phenotype, including a variety of human diseases [7–11]. However, structural variants are also thought to be important for adaptive evolution in natural populations [2, 12–14] and domesticated plants and animals [15], including a number of examples that link structural changes to putative adaptive phenotypic variation [16–20]. From an evolutionary perspective, these larger genomic changes are interesting for a number of reasons. A gene duplication creates a new genetic substrate for evolution to act on, and over long evolutionary timescale, can result in the creation of a paralogous gene [21, 22]. Inversion events can both change the chromatin state that a gene is found in and also suppress recombination events within the inverted region [23, 24]. Finally, structural changes can create incompatibilities between populations, contributing to speciation [14, 25].

For these reasons, it is desirable to understand how genomic rearrangements modify phenotype and spread through populations. However, determining the effect of naturally-occurring genomic rearrangement on phenotype and fitness is very difficult due to linkage of nearby mutations. Experimental evolution is a powerful approach to study adaptation in real time due to the lower rate of nucleotide diversity between the selected strains, aiding in the identification of causal mutations [26–38]. These studies typically utilize microorganisms with short generation times such as *E. coli* or *S. cerevisiae*, elucidating the molecular basis of adaptation and profiling genome dynamics in evolving population under diverse laboratory settings. By identifying and studying causal genetic variants, important insights into beneficial

mutations have been gained, such as their occurrence frequency, the complexity of their molecular basis, the role of contingency and genetic background into their effect, and their fitness effects in specific environments. A number of studies have demonstrated that genomic rearrangements can spread in these populations due to the actions of positive selection [28, 29, 39–44]. In some of these experiments, gene duplicates are thought to facilitate the metabolism or transport of a limiting nutrient due to increased protein product responsible for a rate-limiting step of metabolism.

While these experiments have led to fundamental advances in our understanding of evolution in real time, it is desirable to perform similar experiments in multicellular organisms, with specialized tissues and the ability to respond to their environment using a nervous system. However, long-term adaptation studies are still less advanced in multicellular animals. In our lab, we use the nematode *C. elegans* to study the connection between genotype and phenotype. Compared to other species, *C. elegans* has a high-rate of spontaneous structural mutations, as inferred by their presence in mutation accumulation lines and laboratory strains [41, 45–47]. In general, most of these structural changes are thought to be deleterious; they are purged in populations with higher effective populations sizes [46]. However, spread of copy number variants are also observed in animals carrying deleterious mutations, suggesting that positive selection also acts on copy number variants in certain contexts [41]. Structural changes are also common in wild strains of *C. elegans*, consistent with a role of structural variants being beneficial in certain natural environments [1, 48].

Here, we study two historical laboratory strains of *C. elegans*, called N2 and LSJ2. These two strains share the same hermaphrodite ancestor, which was isolated in 1951 from mushroom compost collected in Bristol, UK (**Fig 1A**). In 1958, descendants of this animal were split into two distinct lineages and cultured in different laboratory conditions. The N2 lineage grew on agar plates seeded with bacteria (standard conditions for a *C. elegans* genetics laboratory). After about two decades growing in this environment, this lineage was cryopreserved in Sydney Brenner's lab and named as N2. After Sydney Brenner introduced *C. elegans* to the genetics research community, N2 became the standard reference strain used across the world [49, 50]. The second lineage was cultured in liquid, axenic media composed of liver and soy peptone extract as a food source for about fifty years before it was cryopreserved and named as LSJ2 [51]. In the time between their separation into two lineages and cryopreservation, approximately 300 mutations arose and fixed in either of the lineages [51]. Previous work has identified six causal mutations of these 300 that confer phenotypic change and competitive advantage in the conditions these mutations arose in [51–56].

In this study, we used recombinant inbred lines (RILs) created using the N2 and LSJ2 strains combined with quantitative trait loci mapping (QTL mapping) to non-biasedly identify any additional mutations between N2 and LSJ2 that confer competitive advantage in standard N2-like laboratory growth conditions. During these experiments, we identified a beneficial, spontaneous, and complex inversion/duplication mutation in the *rcan-1* gene that occurred during the construction of the RILs. This complex genomic rearrangement results in the partial duplication and inversion of five different regions, simultaneously duplicating the *rcan-1* coding region and modifying the upstream promoter regions. While the gene copy number of *rcan-1* is duplicated, the changes in upstream regions result in an overall decrease in *rcan-1* expression. Our work demonstrates how the initial mutational events that create gene duplicates can be complicated, result in unexpected changes in gene expression, and provide fitness increases that will result in its fixation in a population.

# Results

## A N2/LSJ2 recombinant inbred line (RIL$_{hf}$) with increased competitive fitness and exploration behavior than either parental strain

Previously, we developed an assay to estimate the competitive fitness difference between two strains. Briefly, two strains are directly competed against each other in standard laboratory

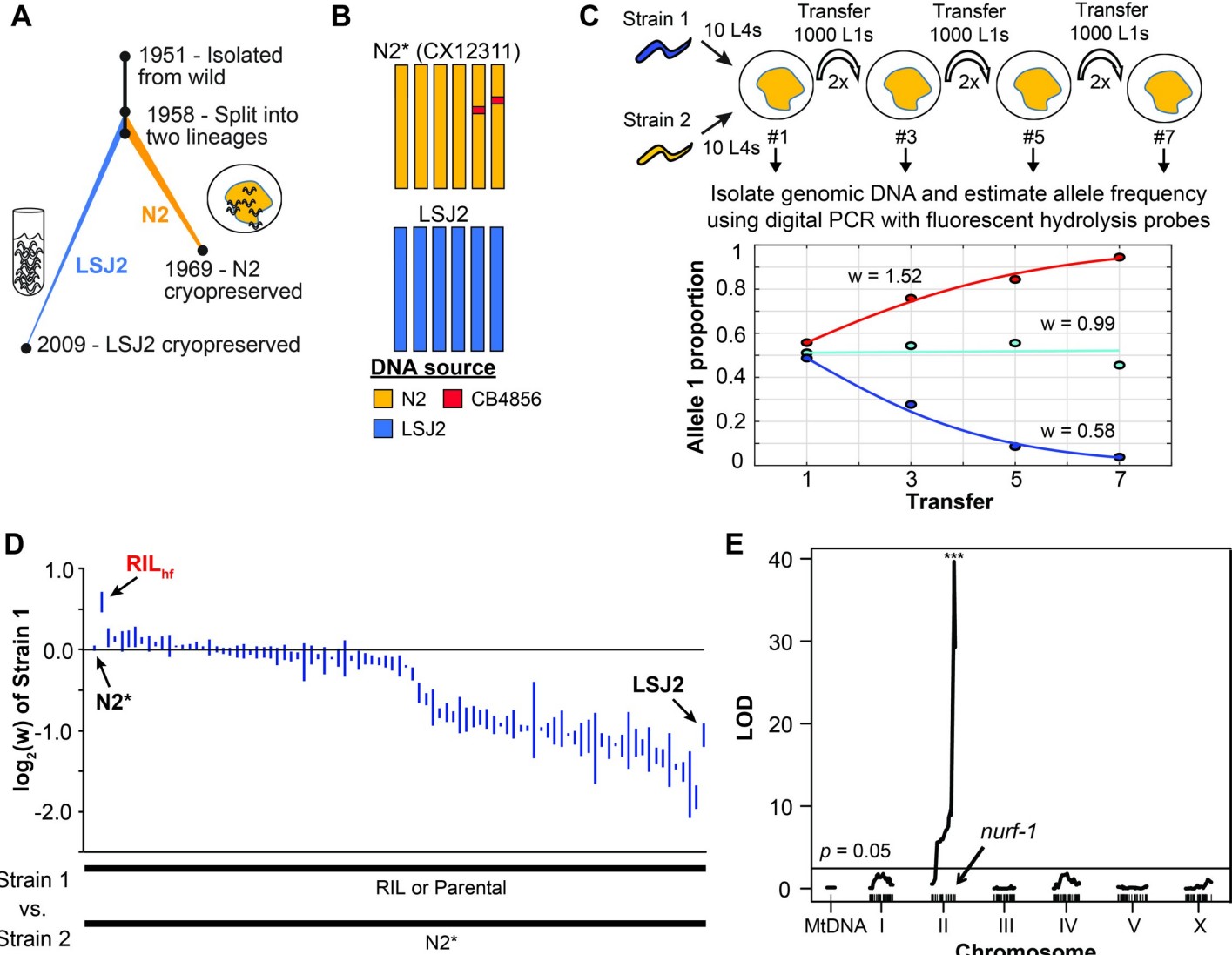

**Fig 1. Competitive fitness measurements of N2*/LSJ2 RILs identifies an outlier RIL.** (A) Overview of the life history of two laboratory strains of *C. elegans* since their isolation from the wild in 1951 and subsequent split into two separate lineages around 1958. The standard reference N2 strain was cultured on agar plates seeded with *E. coli* bacteria until methods of cryopreservation were developed. LSJ2 was cultured in liquid, axenic media until 2009 when a sample of the population was cryopreserved. Resequencing of these strains identified ~300 genetic differences that fixed in one of the two lineages. (B) Schematic of two parental strains used in high-throughput analysis. N2* (or CX12311) is a near-isogenic line (NIL) containing ancestral alleles of two genes, *glb-5* (chromosome V) and *npr-1* (chromosome X) backcrossed from the CB4856 wild strain. Beneficial alleles in these two genes fixed in the N2 lineage; use of the N2* strain allows us to exclude the effects of these alleles from our studies. (C) Example data for three pairwise competition experiments used to quantify the fitness differences between two strains in laboratory conditions. Every odd generation, allele proportion is quantified using digital PCR and fluorescent hydrolysis probes (dots). These points are used to estimate the relative fitness of strain by fitting a haploid selection model to these points (line). In these conditions, outcrossing is expected to be very low or absent due to the lack of males in the initial population. (D) Relative fitness levels were measured for a panel of 89 RIL strains generated between N2* and LSJ2 by competing each RIL against N2* for seven generations. RILs were ordered by their average fitness value (3 replicates were performed for each). Parental strains were also assayed (N2* and LSJ2). RIL$_{hf}$ (red) is highlighted for its unusually high fitness. (E) QTL mapping on the relative fitness differences between the RIL strains. A single significant QTL on the right arm of chromosome II, which overlaps the previously identified *nurf-1* gene, was identified. Threshold line is significance level at *p = 0.05* from a 1,000 permutation test.

growth conditions (i.e. a single agar plate seeded with the OP50 strain of *E. coli* bacteria). Initially, 10 L4 hermaphrodite larva from each strain are transferred to the first plate where they are allowed to eat and reproduce until their grandchildren reach the L1 stage. At this point, ~1000 L1 larva are transferred to a new plate to eat and reproduce until their progeny reach the L1 stage. Subsequently, each generation, ~1000 L1 larva are transferred to a new plate for a total of five to seven generations depending on the experiment. In these conditions, outcrossing is minimized due to the low spontaneous rate of males (confirmed by observing the populations before their transfer to a new plate). The proportion of each strain is then estimated every other generation by isolating genomic DNA from the mixed population and using digital PCR with detection by fluorescent hydrolysis probes targeted to a specific allele pair that distinguishes the two strains. In general, either a naturally-occurring genetic difference between the two strains or a CRISPR-edited silent mutation in the *dpy-10* gene is used (listed in Materials and Methods). Finally, relative fitness is estimated by fitting a haploid model to the measured allele frequencies. This assay is a more direct measure of competitive fitness in laboratory conditions than growth rate, fecundity, or other fitness-proximal traits that are often used in *C. elegans*.

To determine if additional LSJ2/N2 fixed mutations can affect fitness in N2-like laboratory conditions, we used a previously described panel of 89 recombinant inbred lines (RILs) between the CX12311 and LSJ2 strains [51]. CX12311 is a near isogenic line that carries ancestral *npr-1* and *glb-5* alleles from the CB4856 Hawaiian wild strain introgressed into an N2 background (**Fig 1B**—henceforth referred to as N2*). Using N2* as a parental strain eliminates the fitness effect of the derived alleles of N2 *npr-1* and *glb-5* [56]. Using the competition assay described above (**Fig 1C**), we measured the competitive advantage of each of the RIL strains against the N2* strain. A bimodal distribution of relative fitness values was observed in the RIL strains, suggesting that a single genetic locus accounted for the majority of the variation in the RIL strains (**Fig 1D and S1 Table**). Using these measured evolutionary fitness values for QTL mapping, we identified a single significant QTL on the right arm of Chromosome II centered over the *nurf-1* gene (**Fig 1E**). We previously have shown that *nurf-1* contains two fixed mutations from both the N2 and LSJ2 lineages that each affect animal's fitness in N2-like laboratory conditions [54, 57]. These results are consistent with our QTL analysis, suggesting that *nurf-1* plays an important role in adapting to laboratory conditions.

Interestingly, we found that one of the 89 RILs, CX12348 (henceforth called RIL$_{hf}$—hf for high fitness), had significantly higher fitness than either of the LSJ2 or N2* parental strains, which we validated in an independent competition experiment (**Fig 1D** and **Fig 2A**). RIL$_{hf}$ contained a mixture of DNA from both the N2* and LSJ2 parental strains, with N2* DNA on the left arm of chromosome I, the entire chromosomes II, III, and V, and portions of the X chromosome (**Fig 2B**). The higher fitness of RIL$_{hf}$ strain could be caused by two possible reasons: 1) higher-order epistatic interaction between three or more of the 300 derived alleles or 2) a *de novo* beneficial mutation that occurred during construction of the RIL panel. We decided to focus on this unusual strain to determine the genetic basis of its higher fitness.

Wild strains of *C. elegans*, as well as the N2* and LSJ2 parental strains, feed in groups on the borders of bacterial lawns, a strategy known as social behavior [52]. While growing the RIL$_{hf}$ strain on standard plates, we noticed that animals had a stronger propensity to explore the centers and regions outside of the bacterial lawns, causing an increased number of worms and tracks in the center and outside of the lawn (**Fig 2B**). It has been previously shown that increases in exploration behavior is caused by changes in the relative time *C. elegans* spends in roaming and dwelling states in the presence of $O_2$ and other chemical gradients created by the bacteria [58, 59]. To quantify this behavioral difference, we modified a previously described exploration assay [60] to measure long-term (16hrs) exploration behavior in the presence of

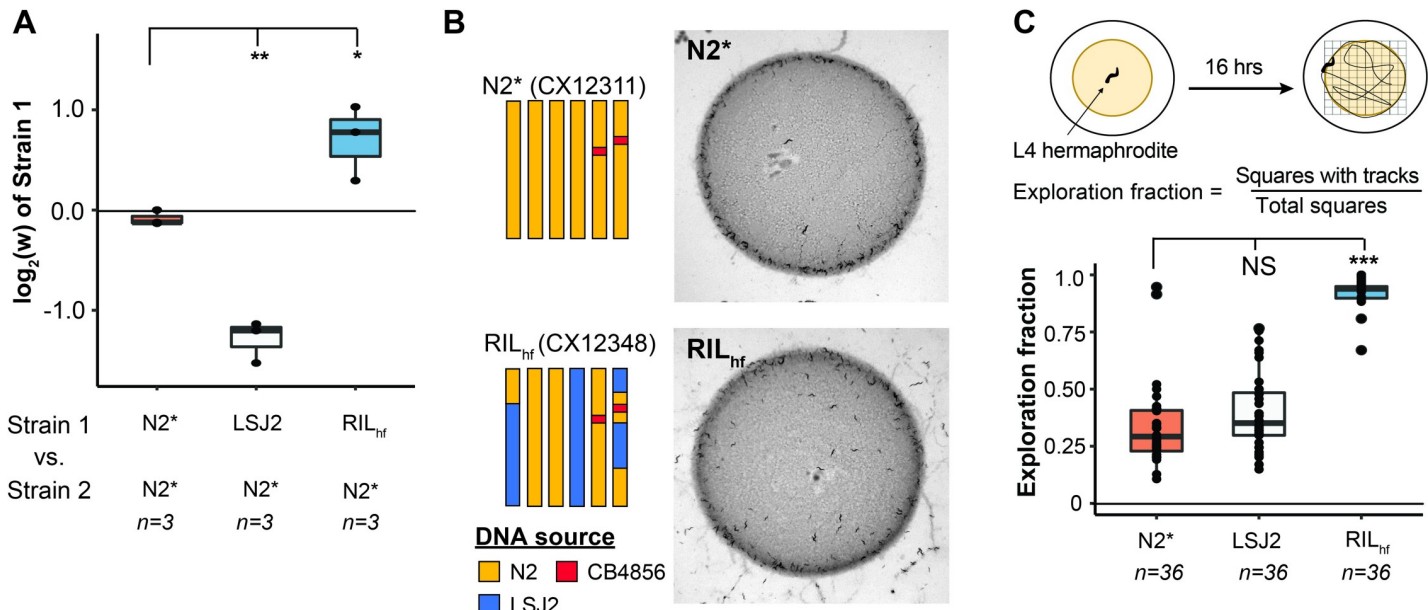

**Fig 2. An outlier RIL with higher pairwise fitness and exploration behavior in laboratory conditions.** (A) The fitness advantage of RIL$_{hf}$ was verified in an independent experiment (*: $p < 0.05$; **: $p < 0.01$—one-way ANOVA tests followed by Tukey's honest significant difference test). (B) Left shows schematic of source DNA of RIL$_{hf}$ (CX12348) from each parental strain. RIL$_{hf}$ contains LSJ2 sequence on chromosome IV and parts of the chromosome I and the chromosome X. RIL$_{hf}$ animals are more likely to be found in the center or outside of the bacterial lawn than at the borders than parental controls (LSJ2 not shown). (C) Exploration behavior differences were quantified by placing a single animal on a plate seeded with a circular lawn. After 16 hours, the amount of the plate that was explored was quantified by counting the number of grid squares with animal tracks within it. Each point represents data from a single animal. The RIL$_{hf}$ explored more of the plate than either parental strain (NS: not significant; ***: $p < 0.001$. one-way ANOVA tests followed by Tukey's honest significant difference test).

circular lawns (instead of uniform lawns used in standard assays). The RIL$_{hf}$ strain explored a substantially larger fraction of the bacterial lawn than either of the parental strains (**Fig 2C**).

## Mapping the causal mutation responsible for higher exploration behavior in RIL$_{hf}$

The change in exploration behavior is potentially an adaptive strategy for RIL$_{hf}$ animals to increase their evolutionary fitness in the laboratory or it might be a pleiotropic effect of the underlying genetic basis of this fitness gain. Since this exploration trait is easier to assay than relative fitness, we first focused on mapping this phenotype using a bulk-segregant approach. We created two new small panels of 48 RILs between RIL$_{hf}$ and either the N2* or LSJ2 parental strains and measured their exploration behavior (**Fig 3A and 3B**). An approximately equal number of RIL strains showed each parental phenotype, suggesting that this trait was controlled by a single locus. We grouped these strains into low or high exploration groups for each RIL panel and performed pooled genomic sequencing (~60x coverage) on the four groups that were created (**Fig 3C**). For each group, we estimated the allele frequency of each of the ~300 N2/LSJ2 genetic variants across the genome. In bulk-segregant analysis, the genetic loci which are not responsible for the exploration behavioral difference are expected to have approximately equal N2/LSJ2 allele proportion. The genetic loci that contribute to exploration behavioral difference are expected to show a larger difference pattern of N2/LSJ2 allele proportion. By analyzing the sequencing result, a large allelic imbalance between the pooled sequencing groups was observed in the center of chromosome III in the RIL$_{hf}$ x LSJ2 panels (**Fig 3D**). This result is expected if a *de novo* mutation arose and fixed in the RIL$_{hf}$ strain in the center of

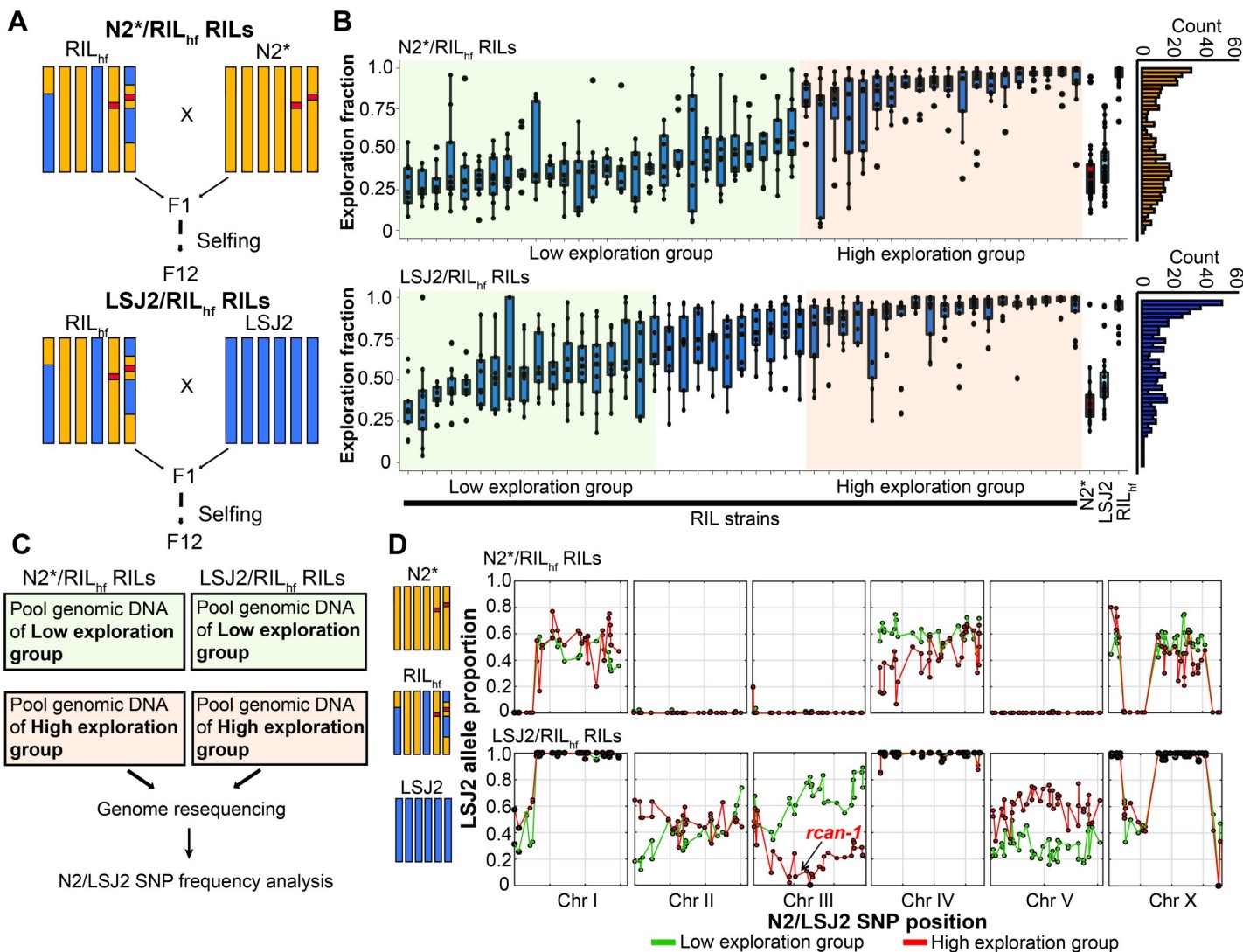

**Fig 3. Exploration behavior differences of the RIL_hf strain maps to the center of chromosome III.** (A) To map the changes in RIL_hf exploration behavior and fitness, we generated two panels of RILs (n = 48) between the RIL_hf and N2* strains or the RIL_hf and LSJ2 strains. (B) Each RIL was measured for exploration behavior using the assay shown in **Fig 2C**. Color coding shows how strains were combined into low (green) or high (orange) groups for bulk-segregant analysis. Uncolored RILs were not included. The histograms on the right displayed the distribution of RIL's exploratory fraction. (9 replicates were performed for each RIL). (C) Overview of bulk-segregant approach using pooled genomic DNA to calculate LSJ2/N2 allele frequency. (D) Allele frequency for LSJ2/N2 genetic differences was calculated for each population. A large allelic frequency difference was observed on chromosomes III and V in the LSJ2/RIL_hf.

chromosome III (which contains the N2 haplotype for the entire chromosome). In this scenario, a similar imbalance would occur for the *de novo*, causal variant in the RIL_hf x N2* cross, however, because the two strains are largely identical on chromosome III, we could not observe it using the LSJ2/N2 SNVs. In addition to the center region of chromosome III, we also detected a large allele frequency difference on the center of chromosome V, suggesting that genetic variation on chromosome V also contributes to exploration behavior. However, the allelic imbalance on V was opposite as our expectation (i.e. the higher exploration group contained LSJ2 alleles on V while the RIL_hf strain contains N2 alleles on V). Since the chromosome III shows a stronger allelic imbalance signal than V (max = ~0.8 vs ~0.6) and goes in the expected direction, we focused on identifying the causal genetic variation in this region.

However, it is possible that variation on chromosome V also contributes to exploration, although potentially in a manner unrelated to the $RIL_{hf}$ phenotype.

## A *de novo* complex genomic rearrangement is identified in the *rcan-1* gene

To determine if the $RIL_{hf}$ strain contains any *de novo* mutations in the center region of chromosome III, we sequenced genomic DNA isolated from the $RIL_{hf}$, N2*, and LSJ2 strains using Illumina short read sequencing. Although we did not identify any *de novo* SNVs or small indels on chromosome III in the $RIL_{hf}$ strain, we did identify a large increase in coverage (2x – 8x) in the *rcan-1* gene, which is an ortholog of human Down Syndrome gene *RCAN1* (**Fig 4A**) [61]. This coverage increase was detected in the high exploration groups of both RIL panels, consistent with this genetic change causing increased exploration behavior (**Fig 4A**). The increased sequencing coverage suggests that the *rcan-1* gene region has been amplified in the $RIL_{hf}$ strain.

While a simple gene duplication event would cause an increase in coverage, we observed a non-uniform change in coverage across the affected region. We also identified a large number of chimeric or split reads (reads which partially align to two unique locations) that mapped to multiple locations within the *rcan-1* locus (**S1 Fig**). The sequence of these chimeric reads within these groups were consistent with each other, and suggest that at least five new fusions between DNA sequence has occurred in the *rcan-1* region of the $RIL_{hf}$ strain. In other words, the *rcan-1* genetic change consists of multiple inversion and/or duplication events. To resolve the precise mutation, we first attempted to amplify the entire affected region using PCR without success. As a complementary approach, we sequenced the $RIL_{hf}$ strain using an Oxford Nanopore sequencing MinION, a long-read single molecule sequencing device with reported read lengths that could resolve the complex rearrangement [62]. By selecting reads that mapped to the *rcan-1* region, we identified a single, ~34.5 kb long read that spanned the entire *rcan-1* region (**Fig 4B** and **S1 Data**). This read resolved the large structural changes of this complex genomic rearrangement. By combining this long Nanopore long read with the DNA fusion events predicted by the Illumina short read sequencing, we resolved the complex rearrangement into five unique tandem inversions interspaced within the *rcan-1* locus (**Fig 4C** and **S2**–**S4 Data**). This proposed rearrangement was consistent with other Oxford Nanopore reads that did not entirely span the rearrangement and resolved the coverage increase and chimeric reads from the Illumina short-read resequencing (**S2 Fig** and **S3 Fig**) as well as smaller PCR products that cover the new junctions (**S4 Fig** and **S2 Table**).

## *rcan-1* complex genomic rearrangement is linked to changes in fitness and exploration behavior

To determine if this rearrangement was responsible for the increases in exploration behavior and relative fitness of the $RIL_{hf}$ strain, we created two near isogenic lines (NILs) by backcrossing the *rcan-1* rearrangement from the $RIL_{hf}$ strain into the N2* background (**Fig 5A**). Genomic DNA from these NILs was sequenced to confirm that LSJ2-derived DNA and $RIL_{hf}$-specific mutations besides the rearrangement were removed from both NILs (**S3 Table**). As expected, both of these NILs explored a higher fraction of the bacterial lawn (**Fig 5B**). Pairwise competition experiments between the NILs and the N2* strain also demonstrated that this rearrangement is associated with the increases in fitness (**Fig 5C**). Finally, we were interested in whether the rearrangement affected fitness-proximal traits such as body size, growth rate, or reproduction. We used a high-throughput COPAs worm sorter to demonstrate that the NIL animals were shorter than wild type controls (**Fig 5D**), indicating that at least one fitness-proximal trait (body length) was affected. However, we cannot say whether this difference in body

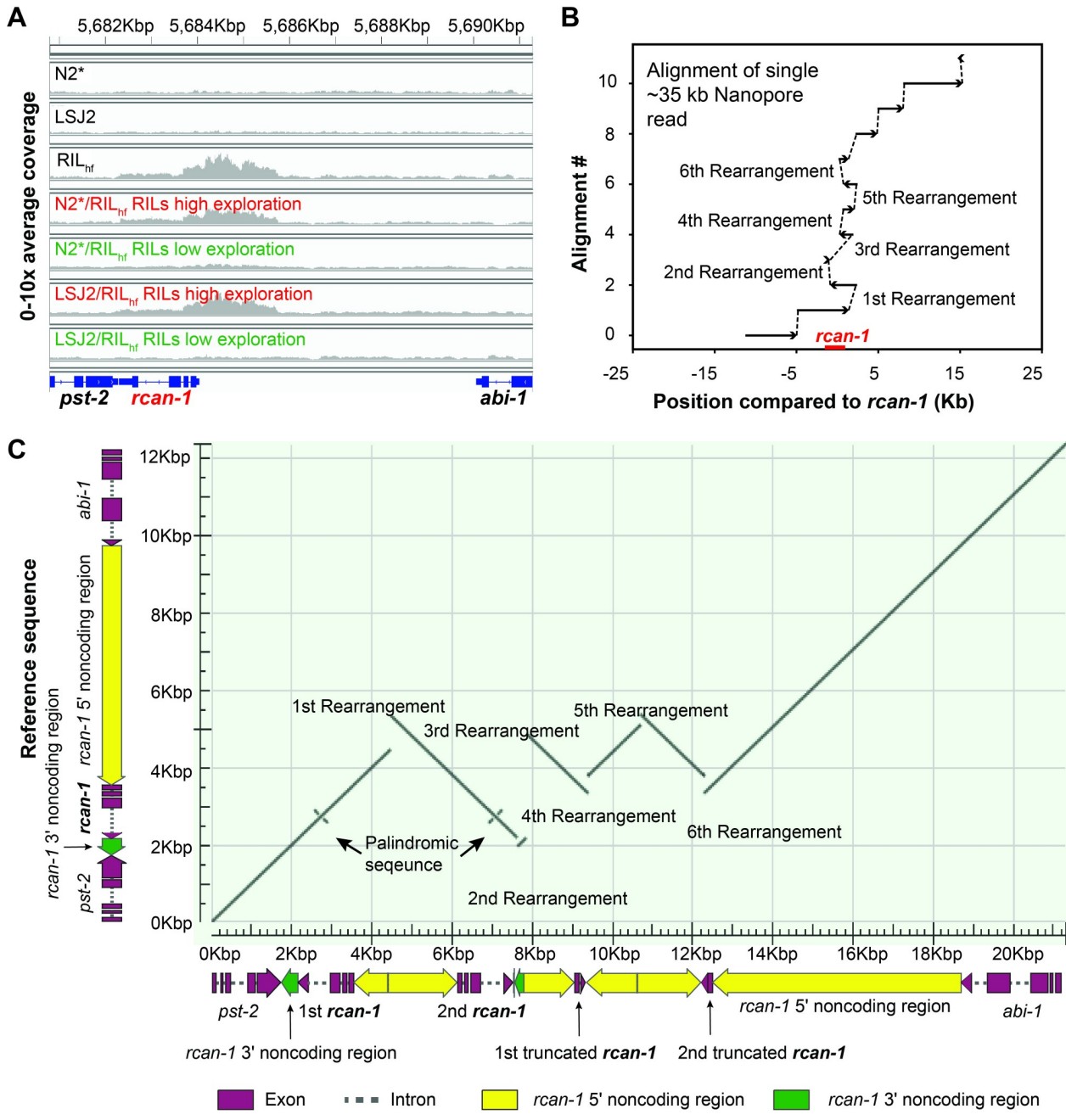

**Fig 4. A *de novo*, complex rearrangement in *rcan-1* in the RIL_hf strain.** (A) Illumina resequencing of the RIL_hf strain identified an increase in coverage at the *rcan-1* locus that was not present in either the N2* or LSJ2 parental strains. This increase in coverage was linked with high exploration in both RIL panels, consistent with a role in exploration behavior. We were unable to resolve the exact nature of the genetic change using the short reads. (B) A single ~34.5 kb read from an Oxford Nanopore Minion resolved the *rcan-1* rearrangement. This read was aligned to the N2 reference using blastn. Alignments are numbered on the y axis. Alignment gaps can be caused by either poor sequence quality of the read, or by genomic rearrangements in the RIL_hf strain. The x-axis shows the position of the read relative to *rcan-1*. To resolve the junctions, we used chimeric reads from the Illumina resequencing in (A). (C) Dot plot of the *rcan-1* rearrangement. The y-axis shows the reference sequence of *rcan-1*, and the x-axis shows the rearrangement in the RIL_hf strain. A total of six new junctions was observed, causing changes to the *rcan-1* locus shown under the x-axis. Palindromic sequences at the 3rd intron of *rcan-1* gene body are also shown.

length is responsible for the change in fitness. These data support a causal role for the *rcan-1* rearrangement for both the competitive fitness advantage and the exploration behavioral

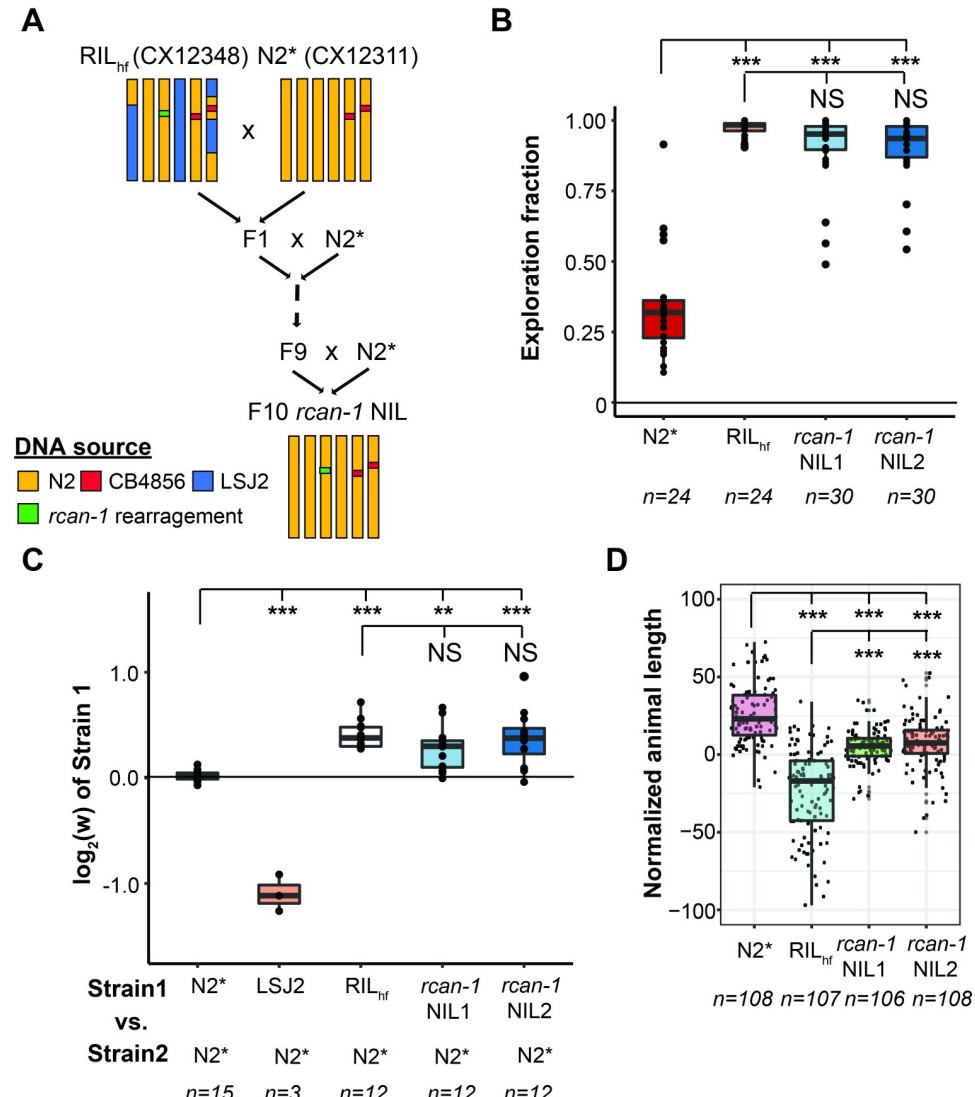

**Fig 5. The *rcan-1* rearrangement is linked to the exploration and fitness differences of the RIL<sub>hf</sub>.** (A) A schematic of the pedigree used to create two near isogenic lines (NIL) by backcrossing the RIL$_{hf}$ strain to the N2* strain. (B) The two *rcan-1* NIL strains showed a similar exploration fraction as the RIL$_{hf}$ strain. (NS: Not significant; ***: $p < 0.001$. one-way ANOVA tests followed by Tukey's honest significant difference test). (C) The relative fitness differences of the NIL strains are comparable to the RIL$_{hf}$ strain. Strains are shown on the x-axis, and the relative fitness of strain 1 is shown on the y-axis. (NS: Not significant; **: $p < 0.01$; ***: $p < 0.001$. one-way ANOVA tests followed by Tukey's honest significant difference test). (D) A high-throughput development assay was used to measure animal lengths for the N2*, RIL$_{hf}$, and two NIL strains. Each point is a biological replicate, with the y-axis indicating the normalized median length of a population of animals. Animal length (μm) measurements are normalized by regressing out the differences among experiments (see Materials and Methods). The lengths of both NILs are significantly different from the lengths of the N2* and RIL$_{hf}$ strains (***: $p < 0.001$. one-way ANOVA tests followed by Tukey's honest significant difference test).

changes of the RIL$_{hf}$ strain. However, we do not exclude a role for additional genetic mutations in regulating these phenotypes, such as genetic variation on chromosome V suggested by the bulk-segregant analysis of exploration behavior.

The *rcan-1* rearrangement is predicted to cause a number of changes to the *rcan-1* gene. First, it creates two full-length versions of the *rcan-1* coding region (Fig 4C). However, the upstream region for each is modified by an inversion event in between the two coding regions.

The first and second copies of *rcan-1* contain 857 and 1,725 bp of endogenous upstream sequence before the inversion event occurs. While the core promoter region is likely conserved in both of the full-length *rcan-1* versions, enhancers and other regulatory regions are probably missing or perturbed in the rearranged region, which might cause decreased, increased, or ectopic expression of *rcan-1* (e.g. in *C. elegans*, at least 3kb of upstream DNA is typically used to estimate the expression pattern of a given gene). Indeed, analysis of previously published ChIP-seq data [63–65] identified 39 transcription factors that bind throughout the upstream region of *rcan-1* (S5 Fig and S4 Table). Second, the second copy of the *rcan-1* gene also contains a small inversion in the 3' UTR region. This inversion could modify binding sites for small RNAs or other RNA-binding proteins that regulate the stability or translation of the mRNA product. Additionally, the 3' end of the small inversion is fused to an upstream promoter region, consequently, the native transcriptional terminator is missing from the second full-length copy of *rcan-1*. Finally, two truncated copies of the *rcan-1* gene are also created, containing the first two exons of the gene. It is possible that truncated peptides with novel C-terminal fragments are produced from these copies and modify wildtype phenotype, although they lack the PxIxIT motif encoded by the last exon of *rcan-1* that is required for RCAN-1 to bind with Calcineurin/TAX-6 [66]. It is difficult to predict *a priori* which of these changes alone or in combination could cause the changes to exploration behavior and/or evolutionary fitness.

## The rearrangement of *rcan-1* decreases its expression but is not a loss-of-function allele

To gain insights into the transcriptional changes caused by the rearrangement, we used RNA-seq to compare the genome-wide expression differences between the two NIL strains and the N2* strain. Interestingly, the gene with the largest change in expression was *rcan-1*, indicating that the rearrangement decreased transcription of the *rcan-1* gene by about 75% (Fig 6A, S6 Fig, and S5 Table). In other contexts, gene duplications can modify phenotype by increasing gene dosage and expression; for the *rcan-1* rearrangement, this is not the case.

Because the transcriptional profiling only reports whole-body changes in total *rcan-1* expression, we created fusions of fluorescent proteins to the modified upstream regions of the different *rcan-1* versions. We cloned the entire region between the two full-length versions of *rcan-1* in both directions to create a P$_{rcan-1-R1}$::mCherry construct (reporting expression of the first full-length version of *rcan-1* in the complex rearrangement) and a P$_{rcan-1-R2}$::mCherry construct (reporting expression of the second full-length version of *rcan-1* in the complex rearrangement). As a control, we also cloned the first 5,085 bp of the upstream region from N2 and fused it to both GFP and mCherry (P$_{rcan-1-WT}$::GFP or P$_{rcan-1-WT}$::mCherry). We then simultaneously co-injected P$_{rcan-1-WT}$::GFP with P$_{rcan-1-WT}$::mCherry, P$_{rcan-1-R1}$::mCherry, or P$_{rcan-1-R2}$::mCherry (Fig 6B). Using a microfluidic device combined with confocal microscopy, we imaged whole-body expression from both green and red channels (Fig 6C). As expected from a previous publication, we observed wild-type expression of *rcan-1* in a variety of tissues, including neurons, pharyngeal cells, and hypodermal cells [61]. We first measured how the modified upstream regions affected whole-body expression by measuring the total amounts of GFP and mCherry signals from ~30 animals for each promoter construct (Fig 6B). Both of the constructs from the complex rearrangement drove less mCherry expression than the wild-type construct to different extents, with the first upstream rearrangement more affected. The effect of the different constructs on mCherry fluorescence levels was also tissue-specific, which we measured by quantifying the appropriate anatomical regions. For example, the head fluorescence was significantly more affected then the body fluorescence in both constructs (Fig 6B).

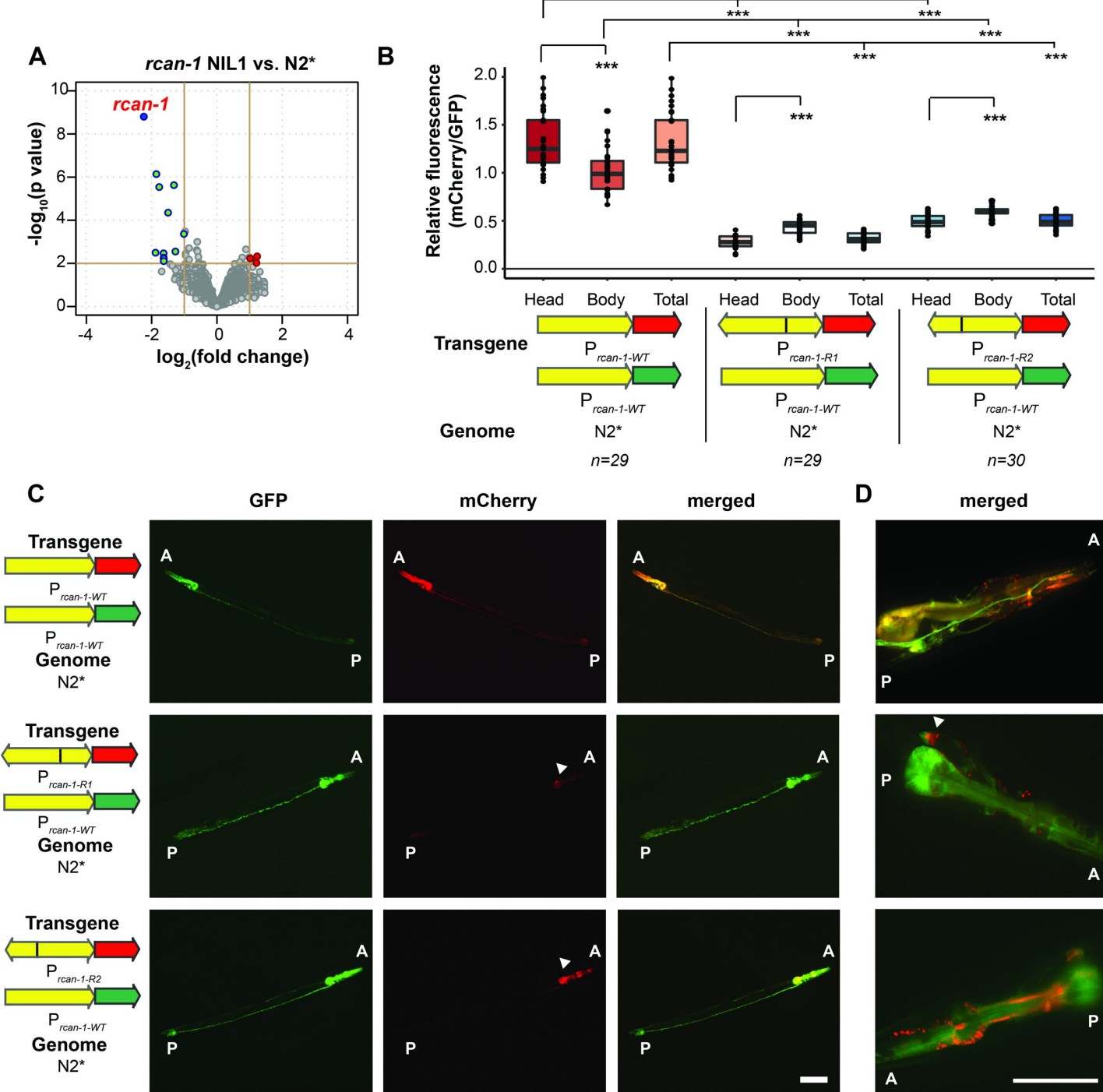

**Fig 6. The *rcan-1* rearrangement allele decreases expression of *rcan-1*.** (A) A volcano plot of expression differences between the *rcan-1* NIL1 and N2* strains. RNA was isolated from synchronized, L4 animals. The gene with the largest and most significant expression decrease was *rcan-1*. Red: p<0.01, log$_2$(Fold Change) > 1. Cyan: p<0.01, log$_2$(Fold Change) < -1. (The list of differential expressed genes with significance are available in **S5 Table**). (B) Co-injection of wild-type *rcan-1* promotors driving GFP with wild-type *rcan-1* or rearranged *rcan-1* upstream promoter regions driving mCherry created from the RIL$_{hf}$ strain. Each dot represents the ratio of total GFP expression divided by total mCherry expression from a single animal. Fluorescence was also segmented into head or body expression and compared separately (***: $p < 0.001$. one-way ANOVA tests followed by Tukey's honest significant difference test. The top comparison group refers to the head fluorescence signal of the two truncated *rcan-1* promoters compared to head of wild-type *rcan-1*). (C) Representative fluorescence images of (B). The white arrow in the middle panel indicates the neurons that retain high level of expression of mCherry driven by P$_{rcan-1-R1}$. The white arrow in the bottom panel indicates the expression of mCherry driven by P$_{rcan-1-R2}$ in the head region. Scale bar is 100μm. (A: anterior; P: posterior). (D) Representative fluorescence images of the head showing a pair of interneurons with less affected expression. (These are different animals from C). The white arrow in the middle panel indicates the neurons that retain high level of expression of mCherry driven by P$_{rcan-1-R1}$. Scale bar is 50μm. (A: anterior; P: posterior).

Further, while the transcriptional reporter P*rcan-1-R1* shows decreased fluorescence in the pharynx, fluorescence in two neurons are mostly unaffected. The cell bodies of these neurons are found in the retrovesicular ganglion and send a single process to the nerve cord. We tentatively identified these neurons as RIF or RIG. The reporter P*rcan-1-R2* universally decreased the expression in the head (**Fig 6D**). Combined with the whole-body RNA-seq data, we suggest that *rcan-1* expression is largely reduced in the RIL$_{hf}$ strain because of changes to the upstream regions of the new versions of *rcan-1*. Potentially, there are cell-type specific changes in transcription, however, extrachromosomal arrays are composed of dozens to hundreds of copies of the promoter region that might not reflect the expression of the genomic promoter.

The above experiments suggest that the *rcan-1* rearrangement could be beneficial because of a global reduction of *rcan-1* transcription. However, an alternative hypothesis is simply that the rearrangement is beneficial because loss of *rcan-1* activity is beneficial in laboratory conditions and the remaining residual expression is unrelated to the fitness of the animals. To test the second hypothesis, we used CRISPR-enabled genomic editing to delete *rcan-1* in the N2* strain (**S7 Fig**). This knockout strain showed an intermediate phenotype between the N2* and the *rcan-1* NIL strains in the modified exploration behavior (**Fig 7A**). When we competed this strain against the two *rcan-1* NILs, we found that the *rcan-1* rearrangement was substantially more fit than the *rcan-1* deletion (**Fig 7B**). We also competed the strain containing the *rcan-1* deletion against wild-type N2* and found no significant difference in fitness (**Fig 7C**). These data are consistent with the rearrangement producing residual or ectopic expression of *rcan-1* that is necessary for the fitness gains. We attempted to rescue the RIL$_{hf}$ exploration phenotype using a transgene created from a PCR product amplified from the wildtype *rcan-1* region, however, this construct was unable to rescue the exploration behavior (**S8 Fig**). There are a number of putative explanations for this result. Potentially, the transgene does not fully recapitulate wildtype expression of *rcan-1*, lacking upstream elements required for expression in cells necessary for the changes in the exploration behavior. Alternatively, additional genetic variants in RIL$_{hf}$ also promote exploration behavior independent of the *rcan-1* rearrangement.

While the changes in exploration behavior are genetically linked to the changes in fitness on laboratory plates, it is unknown whether the changes in exploration behavior are required for the fitness gains. To test this, we used plates seeded with a uniform bacteria lawn (UBL) across the entire plate. These plates lack the lawn borders that create $O_2$ gradients and suppress aggregation behavior of N2* animals [56]. We competed RIL$_{hf}$ against N2* on UBL plates and found that RIL$_{hf}$ still showed a fitness advantage (**Fig 7D**), indicating that the behavioral change is not solely responsible for the fitness gains. We also tested whether the RIL$_{hf}$ strain consumed more food than the N2* strain. We had previously found that a strain containing derived, beneficial alleles of *npr-1* and *glb-5* consumed more food on plates in an equal amount of time. However, the food consumption of the RIL$_{hf}$ strain was statistically indistinguishable than the N2* strain (**S9 Fig**).

Finally, in order to explore the role between gene dosage of *rcan-1* and exploration behavior, we assayed heterozygotes between the RIL$_{hf}$, NIL and *rcan-1* deletion strains (**Fig 7E**). These experiments suggest that there is a strong relationship between *rcan-1* dosage and exploration behavior, as the heterozygotes for each of these crosses were intermediate to the parental strains.

## Discussion

In this study, we identified an outlier RIL with higher relative fitness than either parental strain. This RIL also displayed a new behavioral phenotype not seen in either parental strain, resulting in increased exploration activity on laboratory agar plates. By mapping this trait, we

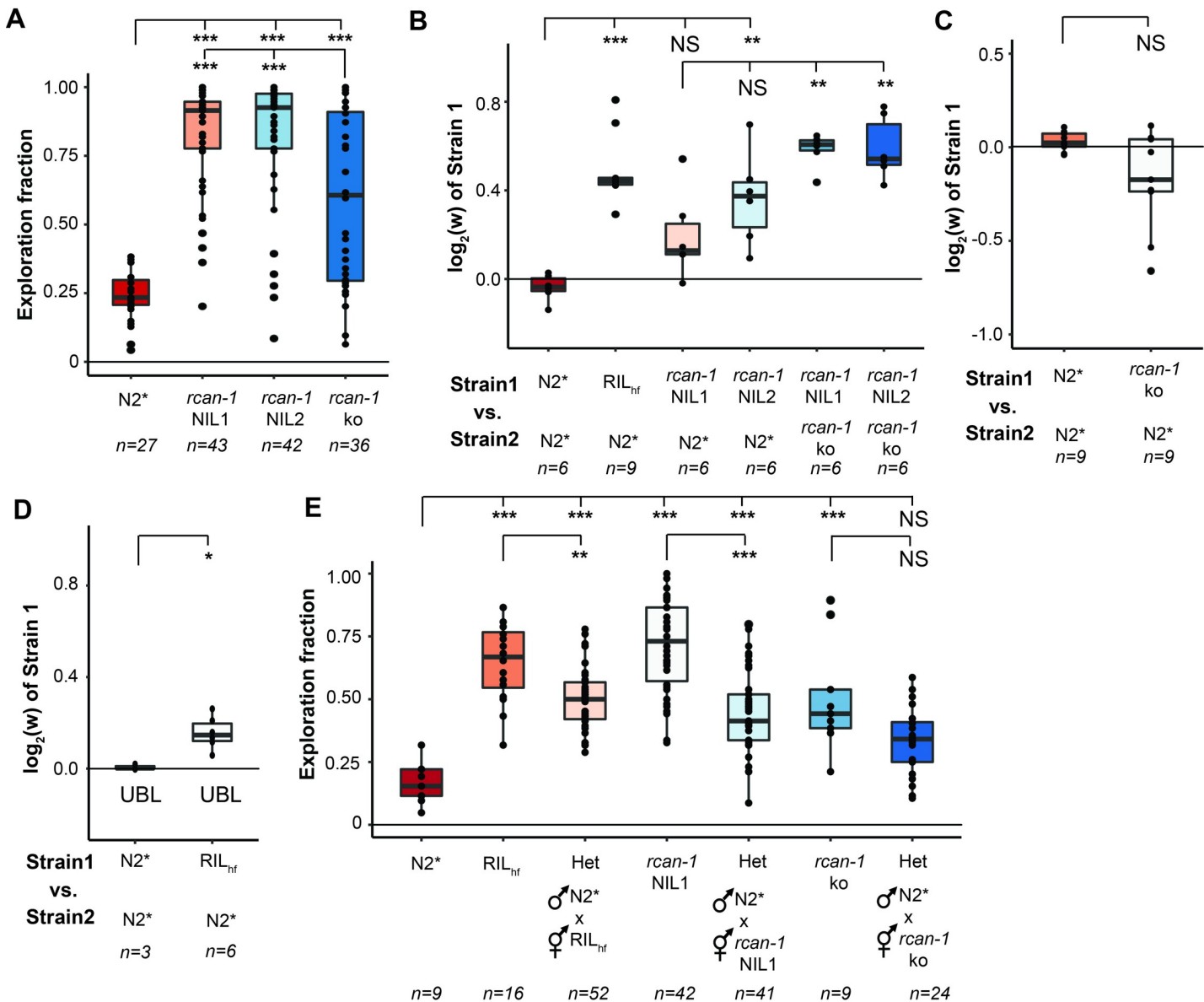

**Fig 7. The *rcan-1* rearrangement allele is not a loss of function allele but its complexity is necessary for fitness advantage and active exploration behavior.** (A) A large deletion of the *rcan-1* coding region was created using CRISPR/Cas9 genomic editing of the N2* strain. The *rcan-1* knockout modified exploration behavior but did not phenocopy the *rcan-1* NIL strains (***: $p < 0.001$. one-way ANOVA tests followed by Tukey's honest significant difference test). (B) Competition experiments demonstrated that a strain carrying an *rcan-1* deletion allele was less fit than the *rcan-1* NIL strain (NS: Not significant; **: $p < 0.01$; ***: $p < 0.001$. one-way ANOVA tests followed by Tukey's honest significant difference test). (C) Competition experiments suggested that a strain carrying an *rcan-1* deletion allele does not show fitness advantage when compete against *rcan-1* wild-type strain. (NS: Not significant. Unpaired Mann-Whitney-Wilcoxon Test). (D) Plates seeded with uniform bacteria lawn that suppress aggregation behavior of N2* do not fully suppress fitness advantage of *rcan-1* rearrangement allele. (*: $p < 0.05$. Unpaired Mann-Whitney-Wilcoxon Test). (E) F1 heterozygotes N2* x RIL_hf animals and N2* x *rcan-1* NIL1 animals show significantly lower exploration fraction than RIL_hf and *rcan-1* NIL1. (NS: Not significant; **: $p < 0.01$; ***: $p < 0.001$. one-way ANOVA tests followed by Tukey's honest significant difference test).

identified a tandem set of inversion/duplications in the *rcan-1* gene that seemed to influence both the exploration behavior and relative fitness of this RIL in standard laboratory conditions. A complex genomic rearrangement affecting phenotype have been found in *C. elegans* before [67], however, this was created in response to a chemical mutagen. The genomic rearrangement of *rcan-1* was unexpectedly complex and provides insight into how gene duplication and rearrangement can occur in microevolutionary timescales.

Gene duplicates are thought to be a primary source of genetic material for the generation of evolutionary novelty, however, it is unclear how duplicates can arise and then navigate an evolutionary trajectory from redundancy to a state where both copies are maintained by natural selection as paralogs [68]. Two major issues in understanding how new gene copies evolve are understanding how gene duplicates initially spread through a population and the evolutionary forces responsible for functional differences in the two copies. Our work here suggests how both can occur due to a single mutational event. While some models of gene duplication have focused on the role of masking deleterious mutations or the role of genetic drift and purifying selection in spreading gene duplicates in a population [69, 70], our results suggest that positive selection can also be involved. After isolation from the wild, evolutionary mismatch between *C. elegans* and its laboratory environment resulted in N2* being at a point away from an adaptive peak. One route to increase its fitness was by changing *rcan-1* activity, which was accomplished by a complex genetic change that creates two duplicated copies of the *rcan-1* coding region. This complex genomic rearrangement was created naturally during the short RIL$_{hf}$ construction period (10 generations). We propose this genomic rearrangement occurred as a single genomic instability event, potentially caused by the replication stress or mis-annealing during Okazaki fragment processing in DNA replication. The complex rearrangement might be a unique repair result induced by an initial error that activated a DNA replication checkpoint and the DNA repair machinery [71–73]. Although the RIL$_{hf}$ strain shows a fitness advantage, the breeding pedigree of the RIL panel is designed to minimize fitness effects. We do not propose that the RIL$_{hf}$ was selected for by positive selection despite its high fitness. However, our work demonstrates how the origin of new gene copies can provide a fitness advantage in new environments, where large functional changes to specific genes can be advantageous.

Our work also suggests how functional variation between two gene copies, a second major issue for understanding the evolution of paralogous genes, can arise in a single mutational step. The rearrangement of *rcan-1* causes large-scale changes to upstream noncoding regions, 3'UTR regions, and the creation of two truncated versions of *rcan-1* coding sequence. For short evolutionary timescales, this type of genetic variant could potentially access changes to gene function that would be difficult for a single SNV, insertion-deletion, or tandem duplication to cause. For example, besides changing the upstream promoter region that determines the exact levels and tissues the *rcan-1* gene is expressed in, the complex rearrangement is also predicted to create an *rcan-1* mRNA with a modified 3' UTR, potentially modifying its translational regulation or mRNA stability. Due to the differences in promoter region and 3' UTR, it is possible that the two copies of *rcan-1* are not functionally redundant because they may have different expression levels or tissue-specific expression. It will be interesting to determine the precise amounts of protein that are produced by each copy and whether deleting each of these copies of *rcan-1* has a negative effect on fitness.

Our data indicates that the rearrangement reduces expression of *rcan-1*. Further, analysis of heterozygotes suggests that exploration behavior is sensitive to gene dosage of *rcan-1*. Our working model is that changes in expression of *rcan-1* is responsible for the changes to exploration behavior. Exploration is controlled by a distributed neural circuit [60, 74, 75]. Modifying *rcan-1* activity in these neurons could be responsible for the behavioral changes.

*rcan-1* encodes an ortholog of the human *RCAN1* gene [66], which encodes a calcipressin family protein that inhibits the calcineurin A protein phosphatase [76]. In humans, *RCAN1* plays an important role in human health; it has been proposed to be a key contributor to Down Syndrome phenotypes in patients with trisomy 21 [76, 77] and chronic overexpression of *RCAN1* in mice results in phenotypes related to Alzheimer's disease [78]. In *C. elegans*, *rcan-1* is required for memory of temperature exposure through a *tax-6*/calcineurin-family and *crh-1*/CREB-dependent pathway [66]. Thermotaxis, however, is not predicted to be

important for laboratory fitness, and it is likely that the *rcan-1* rearrangement regulates other unknown aspects of *C. elegans* biology on which selection can act. Unlike the standard N2 strain, which is potentially more fit in laboratory environments due to its ability to consume more food than the N2* strain, a strain containing the *rcan-1* rearrangement showed no difference in food consumption compared to the N2* strain. However, we found that animals that carry the *rcan-1* rearrangement were shorter than the N2* strain. *rcan-1* was previously shown to regulate body size using loss-of-function mutations [79]. It should be interesting to determine the exact phenotypes that are responsible for the gains in fitness in laboratory conditions. While an increasing number of causal genetic variants that modify phenotype and fitness are being identified, few examples demonstrating the exact phenotypes responsible for fitness changes have been worked out.

It will be interesting to study the continued evolution of a strain carrying this rearrangement, as it is unlikely that this strain has reached its adaptive peak in a single mutational step. Will additional beneficial mutations act through *rcan-1*? One possibility is that *cis*-regulatory mutations could fine tune the expression of each copy of *rcan-1* in causal tissues. These mutations could act to further diversify the function of each copy of *rcan-1*. Alternatively, one of the duplicated *rcan-1* copies could be subsequently lost, as seen in experimental evolution of poxviruses [40].

As long-read sequencing technology improves, the ability to identify complex structural variants similar to the one that we described here will increase. It will be interesting to see how often these types of variants survive the actions of purifying and positive selection to become common in natural populations of *C. elegans* and other animals.

## Materials and methods

### *C. elegans* growth conditions

Animals were cultivated on standard nematode growth medium (NGM) plates containing 2% agar seeded with 200 μL of an overnight culture of the *E. coli* strain OP50 [50]. Ambient temperature was controlled using an incubator set at 20°C. Strains were grown for at least three generations without starvation before any experiments were conducted.

### Strains

The following strains were used in this study. For each figure, a list of strains used is included in **S1 Table**.

Near isogenic lines (NILs):

CX12311 (N2*)—*kyIR1(V, CB4856>N2)*, *qgIR1(X, CB4856>N2)*

PTM413 (*rcan-1* NIL 1) *kahIR16(III, CX12348>N2)*, *kyIR1(V, CB4856>N2)*, *qgIR1(X, CB4856>N2)*

PTM414 (*rcan-1* NIL 2), *kahIR17(III, CX12348>N2)*, *kyIR1(V, CB4856>N2)*, *qgIR1(X, CB4856>N2)*

Recombinant inbred lines (RILs):

*CX12311 –LSJ2 RILs*: CX12312-19, CX12321-27, CX12346-52, CX12354-60, CX12362-66, CX12368-75, CX12381-88, CX12414-37, CX12495-99, CX12501-08, CX12510, CX12361

*CX12311—CX12348 (RIL_{hf}) RILs*: PTM378-397, PTM421-434, PTM494-503

*LSJ2—CX12348 (RIL_{hf}) RILs*: PTM435-478

CRISPR-generated knockout and barcoded strains:

PTM505: *dpy-10 (kah83)* II, *rcan-1(kah183)* III, *kyIR1(V, CB4856>N2)*, *qgIR1(X, CB4856>N2)*

PTM288: *kyIR1(V, CB4856>N2)*, *qgIR1(X, CB4856>N2) dpy-10(kah83)II*;

Extrachromosomal array strains:

PTM553 *kyIR1(V, CB4856>N2), qgIR1(X, CB4856>N2), kahEx169[P$_{rcan-1-WT}$::GFP 25ng/ μL; P$_{rcan-1-WT}$::mCherry 25ng/μL] Isolate 1.*

PTM554 *kyIR1(V, CB4856>N2), qgIR1(X, CB4856>N2), kahEx170[P$_{rcan-1-WT}$::GFP 25ng/ μL; P$_{rcan-1-WT}$::mCherry 25ng/μL] Isolate 2.*

PTM555 *kyIR1(V, CB4856>N2), qgIR1(X, CB4856>N2), kahEx171[P$_{rcan-1-WT}$::GFP 25ng/ μL; P$_{rcan-1-WT}$::mCherry 25ng/μL] Isolate 3.*

PTM556 *kyIR1(V, CB4856>N2), qgIR1(X, CB4856>N2), kahEx172[P$_{rcan-1-WT}$::GFP 25ng/ μL; P$_{rcan-1-R2}$::mCherry 25ng/μL] Isolate 1.*

PTM557 *kyIR1(V, CB4856>N2), qgIR1(X, CB4856>N2), kahEx173[P$_{rcan-1-WT}$::GFP 25ng/ μL; P$_{rcan-1-R2}$::mCherry 25ng/μL] Isolate 2.*

PTM558 *kyIR1(V, CB4856>N2), qgIR1(X, CB4856>N2), kahEx174[P$_{rcan-1-WT}$::GFP 25ng/ μL; P$_{rcan-1-R2}$::mCherry 25ng/μL] Isolate 3.*

PTM559 *kyIR1(V, CB4856>N2), qgIR1(X, CB4856>N2), kahEx175[P$_{rcan-1-WT}$::GFP 25ng/ μL; P$_{rcan-1-R1}$::mCherry 25ng/μL] Isolate 1.*

PTM560 *kyIR1(V, CB4856>N2), qgIR1(X, CB4856>N2), kahEx176[P$_{rcan-1-WT}$::GFP 25ng/ μL; P$_{rcan-1-R1}$::mCherry 25ng/μL] Isolate 2.*

PTM561 *kyIR1(V, CB4856>N2), qgIR1(X, CB4856>N2), kahEx177[P$_{rcan-1-WT}$::GFP 25ng/ μL; P$_{rcan-1-R1}$::mCherry 25ng/μL] Isolate 3.*

PTM566 CX12348 *kahEx185[50ng/ul Prcan-1::rcan-1; 45ng/ul pSM;5ng/ul pCFJ90]*

PTM567 CX12348 *kahEx185[50ng/ul Prcan-1::rcan-1; 45ng/ul pSM;5ng/ul pCFJ90]*

## Strain construction

To create the CX12311-CX12348 and LSJ2-CX12348 RILs, CX12311 males or LSJ2 males were crossed to CX12348 hermaphrodites. 96 F2 progeny (48 from CX12311 x CX12348; 48 from LSJ2 x CX12348) were cloned to individual plates and allowed to self-fertilize for 10 generations to create the inbred lines. One RIL line was lost from the LSJ2xCX12348 cross, creating 47 RILs.

To create the *rcan-1* NILs (PTM413 and PTM414), CX12348 animals were backcrossed to CX12311 for 10 generations. Two completely independent sets of crosses were used to create two independent lines. Primers used to identify male animals containing the rearrangement were: 5'—gagacaatactctgatattagacgcacca -3' and 5'–gctgacaccagcaatcattgttca -3'.

To create the *rcan-1* deletion strain (PTM505), two sgRNAs targeting the 5' region of *rcan-1* and two sgRNAs targeting the 3' end of *rcan-1* were created: sgRNA1: 5'-atttggaagatcatctttac-TGG-3'; sgRNA2: 5'-agtgctgatcaatgatccat-TGG-3'; sgRNA3: 5'-cgtggcatttcaattgctga-TGG-3'; sgRNA4: 5'-tcacatggagatgaagggcg-TGG-3'. CoCRISPR [80] was used to simultaneously edit the *dpy-10* and *rcan-1* genes using the following injection mix: 50ng/μL P*eft-3*::Cas9, 10ng/μL *dpy-10* sgRNA, 25ng/μL of each of the four *rcan-1* sgRNAs, and 500nM *dpy-10(cn64)* repair oligonucleotide. This mix was injected into CX12311 animals and Dpy or Rol animals were singled and genotyped using PCR. An animal with the deleted sequence 5'-caatggatcattgatca. . ...cacgcccttcatctccat-3' was identified.

To create the GFP/mCherry extrachromosomal lines (PTM553-PTM561), four constructs were created. P$_{rcan-1-WT}$::GFP was created by amplifying the *rcan-1* promoter from CX12311 genomic DNA using primers 5'-ctgGGCCGGCCtcggttcaaatacctcatgggaca-3' and 5'-ttGGCGCGCCtttttgttgttaacttatagaaaaaatttcagcaacca-3' and cloning it into the pSM-GFP backbone with restriction enzyme sites 5'-*Fse*I and 3'-*Asc*I. To create P$_{rcan-1-WT}$::mCherry, the *rcan-1* promoter was amplified from CX12311 genomic DNA using primers 5'-tcggttcaaatacctcatggg-gaca-3' and 5'- tttttgttgttaacttatagaaaaaatttcagcaacca-3' and a pCFJ90-mCherry backbone was

amplified using primers 5'- atttttctataagttaacaacaaaaaAcaagtttgtacaaaaaagcaggct-3 and 5'-ccatgaggtatttgaaccgaatagcttggcgtaatcatggtcat-3'. The two fragments were assembled using HI-FI assembly (NEB E5520S). To construct the *P*<sub>rcan-1-R1</sub>::mCherry and *P*<sub>rcan-1-R2</sub>::mCherry plasmids, 5'- tttttgttgttaacttatagaaaaaatttcagca-3' and 5'- gaaacgaaacaaggtgggtcc-3' or 5'-tttttgttgttaacttatagaaaaaatttcagca-3' and 5'- agcggacccaccttgtttc-3' were used to amplify the rearranged promoters from CX12348 genomic DNA. These PCR products were cloned into a pCFJ90-mCherry backbone using HI-FI assembly. Concentrations of each plasmid are indicated for each strain in the strain description.

## Competition experiment

Competition experiments were performed as described previously [56]. In the standard assays, 9 cm NGM plates were seeded with 300 **μ**L of an overnight *E. coli* OP50 culture and incubated at room temperature for three days. In the competition experiment using uniform bacteria lawn (UBL), the 9cm UBL plates were made by pouring overnight *E. coli* OP50 culture onto the NGM plate to cover the whole plate. Excess culture on the plate was removed by pouring off and the plates were left at 20˚C overnight for forming uniform bacteria lawn. In each competition experiment, ten L4 larvae from each strain were picked onto a single plate and cultured for five days. Animals were transferred to identically prepared NGM plates and subsequently transferred every four days. Depending on the experiment, five or seven total transfers were performed. For each transfer, animals were washed off the plates using M9 buffer and collected into 1.5 mL centrifuge tubes. The animals were then mixed by inversion and allowed to stand for approximately one minute to settle adult animals. 50 **μ**L of the supernatant containing approximately 1000–2000 L1-L2 animals were seeded onto fresh plates. The remaining animals were concentrated and used for genomic DNA isolation. Genomic DNA was collected every odd generation using a Zymo DNA isolation kit (D4071). To quantify the relative proportion of each strain, a digital PCR assay was performed with custom TaqMan fluorescent-quenching probes (Applied Biosciences). Genomic DNA was digested with *Sac*I or *Eco*RI for 30 min at 37 $^{\mathrm{o}}$C. The digested products were purified using a Zymo DNA cleanup kit (D4064) and diluted to approximately 1–2 ng/μL. Seven TaqMan probes were designed using ABI software that targeted WBVar00051876, WBVar00601322, WBVar00167214, WBVar00601493, WBVar00601538, *dpy-10 (kah82)*, or *tbc-10(kah185)* (S6 Table). Digital PCR assays were performed using a Biorad QX200 digital PCR machine with standard probe absolute quantification protocol. The relative allele proportion was calculated for each DNA sample using the count number of the droplet with fluorescence signal (Eq 1). To calculate the relative fitness of the two strains using three or four measurements of relative allele proportion, we used linear regression to fit this data to a one-locus generic selection model (Eqs 2 and 3), assuming one generation per transfer.

$$P(A)_t = \frac{No.Allele\ A}{No.Allele\ A + No.Allele\ a} \tag{1}$$

$$P(A)_t = \frac{P(A)_0 W_{AA}{}^t}{P(A)_0 W_{AA}{}^t + (1 - P(A)_0) W_{aa}{}^t} \tag{2}$$

$$log(\frac{\frac{P(A)_0}{P(A)_t} - P(A)_0}{1 - P(A)_0}) = (log(\frac{W_{aa}}{W_{AA}}))t \tag{3}$$

The relative fitness value and Taqman assay information for each competition experiment are included in S1 Table.

## Exploration behavioral assay

The exploration assays from Flavell *et al.* were modified to study exploration in the presence of circular lawns [60]. 35 mm Petri dishes were seeded with 150 **μL** OP50 *E. coli* Bacteria for 24 h before the start of the assay. Individual L4 hermaphrodites were placed in the center of the plate and cultivated in 20°C for 16 hours. The plates were placed on a grid that has 100 squares that cover the whole bacteria lawn. To calculate the exploration fraction, the number of full or partial squares that contained animal's tracks out of bacteria lawn border was quantified. The number of full or partial squares that contain the bacteria lawn was also counted (about 94–96 grids). The exploration fraction was calculated (Eq 4).

$$Exploration\ fraction = \frac{No.\ grids\ contained\ tracks}{No.grids\ contained\ bacteria\ lawn} \tag{4}$$

## Heterozygous exploration assay

Heterozygous F1 was created by mating PTM288 males with the other strain of interest at L4 stage for one day. Fertilized hermaphrodites were then singled into individual plates. After two days, L4 hermaphrodites were picked from plates where a lot of males were present, indicating successful mating. After the assay, these animals are individually lysed and genotyped at *dpy-10(kah83)II* site to confirm they are heterozygous. The genotyping primers: Forward primer: 5'–gtcagatgatctaccggtgtgtcac—3', reverse primer: 5'–gtctctcctggtgctccgtcttcac– 3'.

## *rcan-1* rescue assay

A PCR fragment that covers 4.5kb upstream to 0.7kb downstream of *rcan-1* was cloned using NEB Phusion Q5 PCR system (Forward primer: 5'–gctccatacgcgcatttcag– 3', reverse primer: 5'–tcttctcgaagccgttcacc– 3'). The PCR product was purified and injected at 50ng/uL with 5ng/uL f pCFJ90 and 45ng/uL pSM. The exploration behavior fraction of the animals expressing mCherry was quantified using standard exploration behavior assay method.

## Bulk-segregant analysis of exploration behavior

The exploration behavioral assays were performed on 48 CX12311-CX12348 RILs and 47 LSJ2-CX12348 RILs. In the CX12311/CX12348 RILs, 28 RILs with median exploration fraction less than 0.575 were assigned to the low exploration group and the 20 RILs with median exploration fraction greater than or equal to 0.575 were assigned to the high exploration group. In the LSJ2/CX12348 RILs group, the 17 RILs with median exploration fraction less than 0.620 were assigned to the low exploration group, the 20 RILs with median exploration behavior greater than or equal to 0.870 were assigned to high exploration group, and the rest of the RILs were excluded from further analysis. Genomic DNA from each RIL (100 ng) was isolated and pooled into the four described groups for whole-genome resequencing.

## Whole-genome sequencing

Genomic DNA was isolated using Qiagen Gentra Puregene Kit (158667) following the supplementary protocol for nematodes. The genomic DNA was further purified using Zymo Quick-DNA kit (D4068). DNA libraries were prepared using an Illumina Nextera DNA kit (FC-121-1030) with indexes (FC-121-1011). The prepared libraries were sequenced at 35 bp or 150 bp paired-read using an Illumina NextSeq 500. The reads were aligned to reference genome using BWA-aligner v0.7.17 [81]. BAM files were deduplicated and processed using SAMtools v1.9 [82] and Picard[83] (http://broadinstitute.github.io/picard/). SNVs were called by Freebayes and annotated by SnpEff [84, 85]. Custom Python scripts using the pysam library

([https://github.com/pysam-developers/pysam](https://github.com/pysam-developers/pysam)) were used to identify regions of the genome with a large number of clipped and chimeric reads. Reads depths were visualized using IGV [86]. The sequencing reads were uploaded to the SRA under BioProject PRJNA526525.

## Oxford Nanopore long-read sequencing

Genomic DNA of CX12348 was isolated from animals grown on 8 9 cm NGM plates using Qiagen Gentra Puregene Kit (158667) following the supplementary protocol for nematodes. The genomic DNA was concentrated and purified using Zymo Quick-DNA kit (D4068). Size-selection to collect DNA fragments from 10 kbp– 50 kbp was carried out using a Blue-pippin. The sequencing library was prepared using 1D ligation kit (SQK-LSK108) following the standard protocol. DNA was repaired using the NEBNext FFPE Repair Mix (M6630). After DNA repair, end preparation was performed and the adapter was ligated. 600 ng prepared library was loaded in the Nanopore R9 flow cell in MinION sequencer. The standard 48 hours sequencing protocol was performed and approximately 5 Gb of sequencing data was generated. To resolve the structure of *rcan-1* complex rearrangement, the FASTQ files were aligned to reference genome using BWA aligner. Reads that covered the *rcan-1* gene region and contained a gap in alignment were fetched using pysam ([https://github.com/pysam-developers/pysam](https://github.com/pysam-developers/pysam)). These reads were then mapped to *rcan-1* using BLAST and visualized with matplotlib ([https://matplotlib.org](https://matplotlib.org)) to show the rearrangement events. The structure of the complex rearrangement was verified by using BWA and IGV to map the Illumina short reads or FlexiDot [87] to map and visualize the Oxford Nanopore reads. The sequencing reads were uploaded to the SRA under BioProject PRJNA526525.

## RNA-seq and transcriptome analysis

CX12311, PTM413, and PTM414 were synchronized using alkaline-bleach to isolate embryos, which were washed with M9 buffer and placed on a tube roller overnight. Approximately 400 hatched L1 animals were placed on NGM agar plates for each strain and incubated at 20˚C for 48 hours. The ~L4 stage animals were washed off for standard RNA isolation using Trizol. Four replicates for each strain were performed on different days. The RNA libraries were prepared using the NEB Next Ultra II Directional RNA Library Prep Kit (E7760S) following its standard protocol. The libraries were sequenced by Illumina NextSeq 500. The reads were aligned by HISAT2 using default parameters for pair-end sequencing. Transcript abundance was calculated using HTseq and then used as inputs for the SARTools [88, 89]. edgeR v3.16.5 was used for normalization and differential analysis[55[90]. The analysis result was shown in a volcano plot. CX12311 was treated as the wild type. The genes show significant differential expression in the volcano plot are under thresholds $|\log_2(\text{fold})| > 1$ and FDR adjusted p-value < 0.01. Sequencing reads were uploaded to the SRA under BioProject PRJNA526525.

## Imaging

The detailed steps of microfluidic device fabrication were previously reported [91]. For each experiment, about 100–150 animals were suspended in 1 mL of S Basal and delivered into using a syringe. Animals were immobilized using 1 mL of tetramisole hydrochloride (200 mM) (Sigma-Aldrich cas. 5086-74-8) in S Basal. Imaging were acquired on a spinning disk confocal microscope (PerkinElmer UltraVIEW VoX) with a Hamamatsu FLASH 4 sCMOS camera. Images of the animals were quantified using ImageJ. A region-of-interest (ROI) was drawn around the entire worm, and the mean intensity of the GFP and mCherry images were calculated across the ROI. Relative fluorescence intensity was calculated as (Mean Intensity of mCherry)/(Mean Intensity of GFP).

## Food consumption assay

The experimental method was described previously [56]. In brief, The 24-well plates were prepared by pipetting 0.75mL NGM agar contain 25 μM FUDR and 1x Antibiotic-Antimycotic (ThermoFisher 15240062) to each well. Each well was seeded with 20μL of freshly cultured OD600 of 4.0 (CFU ~ $3.2 \times 10^9$/mL) *E. coli* OP50-GFP(pFPV25.1). The plates were dried in a fume hood and dried with air flow for 1.5hr. The fluorescence signal of OP50-GFP was quantified by area scanning protocol using BioTek Synergy H4 multimode plate reader. The synchronized L4 animals were placed in the wells in the first five columns and the last column is used as control column. Each well was placed with 10 animals, and the plate was incubated in a 20°C incubator for 18 hours and the fluorescence signal was quantified again as the ending time point. The relative food consumption amount was calculated using the equations reported previously [56].

## High-throughput growth rate analysis

The high-throughput growth rate and brood size assays were performed as described previously [92]. In short, approximately 25 bleach-synchronized embryos were aliquoted into each well of 96-well plates, and fed 5 mg/mL HB101 bacterial lysate on the following day [93]. After 48 hours of growing at 20°C, a large-particle flow cytometer (COPAS BIOSORT, Union Biometrica, Holliston, MA) was used to sort three L4 larvae into each well of a 96-well plate with 50 μL of K medium with HB101 lysate (10 mg/mL) and Kanamycin (50 μM). Animals were grown for 96 hours at 20°C and were then treated with sodium azide (50 mM in M9). Animal number (n) and animal length (time of flight, TOF) were measured by the BIOSORT. For each well, animal growth was measured as the median length of the population, and brood size was measured as the number of progeny per sorted animal. The experiments were replicated in two independent assays, and the linear model with the formula (phenotype ~ assay) was applied to normalize the differences among assays [94].

## Statistical test

The raw data are included in **S1 Table**. To assess statistical significance, we performed one-way ANOVA tests followed by Tukey's honest significant difference test to correct for multiple comparisons or the Wilcoxon-Mann-Whitney nonparametric test for pairwise comparisons. NS: not significant; *: $p < 0.05$; **: $p < 0.01$; ***: $p < 0.001$.

## QTL mapping

The average of the $\log_2(w)$ of each N2*/LSJ2 RIL was used as phenotype with 192 previously genotyped SNPs. R/qtl was used to perform a one-dimensional scan using marker regression on the 192 markers. The genome-wide error rate (p = 0.05) was determined by 1000 permutations test[95].

## List of key resources and reagents

The key resources and reagents used in this study are listed in **S6 Table**.

# Supporting information

**S1 Fig. Illumina reads mapped to the *rcan-1* locus.** (A) IGV plot of illumina sequencing short reads align to *rcan-1* genomic locations. (B) Chimeric reads align to *rcan-1* genomic locations. Reads are from the resequencing of the N2*(CX12311), LSJ2, and RIL_hf (CX12348) strain. Besides an increase in coverage at the *rcan-1* locus, a large number of chimeric reads

(i.e. reads that partially map to two locations) were found in the RIL$_{hf}$ strain. (Reads with grey color indicates they are normal reads (Pair orientations: LR); Reads with cyan color imply inversion (Pair orientations: LL); Reads with blue color imply inversion (Pair orientations: RR); Reads with green color imply duplication or translocation (Pair orientations: RL). Reads with red color have larger than expected inferred sizes.)
(TIF)

**S2 Fig. Dot plot of the nanopore sequencing reads align to proposed *rcan-1* rearrangement.** 10 nanopore sequencing reads that overlapped the *rcan-1* structural variant were used to generate a dot plot with proposed *rcan-1* rearrangement.
(TIF)

**S3 Fig. Illumina short sequencing reads aligned to the proposed *rcan-1* structural variant.** Top: All reads aligned to the *rcan-1* rearrangement. Bottom: Chimeric reads aligned to the *rcan-1* rearrangement. The uniform coverage and lack of chimeric reads is consistent with the proposed structure of the rearrangement. (Reads with grey color indicates they are normal reads (Pair orientations: LR); Reads with cyan color imply inversion (Pair orientations: LL); Reads with red color have larger than expected inferred sizes. Reads with empty color have low mapping quality.)
(TIF)

**S4 Fig. The PCR products include the rearranged regions.** Red arrows are the PCR products that include the rearranged regions. The detail information of the primers, the expected length and observed length in agarose gel of each PCR product is listed in **S2 Table**.
(TIF)

**S5 Fig. Transcription factor binding regions at *rcan-1* 5'-UTR.** The green bars represent the transcription factor binding region. The red bars represent the two truncated promoter regions that drive full length of *rcan-1* gene body in the complex rearrangement. The blue bar represents the highly occupied target region ('HOT'). The figure is generated from Wormbase J-browser by adding the feature of transcription factor binding regions. The information of the transcription factors is listed in **S4 Table**.
(TIF)

**S6 Fig. Volcano plot of *rcan-1* NIL2 gene expression vs. N2\*.** Red dots indicate genes with increased expression in *rcan-1* NIL2 vs. N2\* (p<0.01, log2(Fold Change) > 1). Cyan dots indicate genes with decreased expression in *rcan-1* NIL2 vs. N2\* (p<0.01, log2(Fold Change) < -1). The list of differential expressed genes with significance are available in **S5 Table**.
(TIF)

**S7 Fig. Strategy for creating a knockout allele of *rcan-1* using CRISPR/Cas9.** The position of two pairs of sgRNAs that target the 5' and 3' end of the *rcan-1* coding region. The resulting deletion allele is shown as a blue box.
(TIF)

**S8 Fig. Exploration fraction of *rcan-1* rescue lines.** The RIL$_{hf}$ animals were co-injected with 50ng/uL P$_{rcan-1(4.5Kbps)}$::*rcan-1* PCR product, 5ng/uL pCFJ90, and 45ng/uL pSM. The exploration fraction of the animals that express mCherry were measured.
(TIF)

**S9 Fig. Food consumption assay of RIL$_{hf}$ and *rcan-1* NILs.** Relative food consumption of indicated strains. Each dot indicates one experimental replicate.
(TIF)

**S1 Data. *rcan-1*_NanoporeReads.txt.** This file contains the sequence of the Oxford Nanopore reads (**Fig 4B** and **S2 Fig**) that overlap the structural variant in fasta format.
(TXT)

**S2 Data. *rcan-1*_RearrangementSequence.txt.** This file contains sequence information of the proposed *rcan-1* structural variant in fasta format.
(TXT)

**S3 Data. *rcan-1*_RearrangementSequence.txt.** This file contains annotated gene and junction information for the structural variant in Genbank format.
(TXT)

**S4 Data. *rcan-1*_RearrangementSequence.dna.** This file annotated gene and junction information for the structural variant in SnapGene format. It contains the primer information for study the structural variant. This file can be viewed by SnapGene software or SnapGene Viewer software (SnapGene Viewer is a free software).
(DNA)

**S1 Table. Raw data.** This table includes the raw experimental data of **Figs 1–7** and **S8 Fig** and **S9 Fig**.
(XLSX)

**S2 Table. Rearranged junction sequences.** This table includes the junction sequences for the *rcan-1* structural variant. The primer's information and the information of each PCR product's size are also included.
(XLSX)

**S3 Table. NIL resequencing.** This table includes all genetic variants identified in the *rcan-1* near isogenic lines (NILs).
(XLSX)

**S4 Table. TF binding regions in 5 UTR.** This table summarizes the transcription factor binding information at *rcan-1* 5' upstream region from Wormbase.
(XLSX)

**S5 Table. NIL_RNA-Seq.** This table includes all gene expression data for *rcan-1* NILs.
(XLSX)

**S6 Table. Sequence information of TaqMan probes and summary of resources and reagents.** This table lists sequence information for the TaqMan fluorescent quenching probes used for competition experiments. This table also includes the information of key resources and reagents used in this study.
(XLSX)

## Acknowledgments

We thank the *Caenorhabditis* Genetics Center for strains, Todd Streelman, Levi Morran, Chao Jiang, Wei Zhang, Will Ratcliff, Annalise Paaby, and members of the Streelman and McGrath lab for discussions, and WormBase.

## Author Contributions

**Conceptualization:** Yuehui Zhao, Patrick T. McGrath.

**Data curation:** Yuehui Zhao, Lijiang Long, Jason Wan, Shannon C. Brady, Daehan Lee, Akinade Ojemakinde, Fredrik O. Vannberg, Hang Lu.

**Formal analysis:** Yuehui Zhao, Lijiang Long, Shweta Biliya, Erik C. Andersen, Patrick T. McGrath.

**Funding acquisition:** Erik C. Andersen, Patrick T. McGrath.

**Investigation:** Lijiang Long, Shweta Biliya, Patrick T. McGrath.

**Methodology:** Lijiang Long, Jason Wan, Fredrik O. Vannberg, Hang Lu.

**Project administration:** Patrick T. McGrath.

**Resources:** Fredrik O. Vannberg.

**Supervision:** Erik C. Andersen, Hang Lu.

**Validation:** Yuehui Zhao, Patrick T. McGrath.

**Visualization:** Yuehui Zhao, Lijiang Long, Jason Wan, Shannon C. Brady, Daehan Lee, Patrick T. McGrath.

**Writing – original draft:** Yuehui Zhao, Patrick T. McGrath.

**Writing – review & editing:** Yuehui Zhao, Erik C. Andersen, Patrick T. McGrath.

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
