## [Decision Letter · Decision Letter 0]

19 Sep 2019

Dear Dr McGrath,

Thank you very much for submitting your Research Article entitled 'A spontaneous complex structural variant in rcan-1 increases exploratory behavior and laboratory fitness of C. elegans' to PLOS Genetics. Your manuscript was fully evaluated at the editorial level and by independent peer reviewers. The reviewers appreciated the attention to an important topic but identified some aspects of the manuscript that should be improved.

We therefore ask you to modify the manuscript according to the review recommendations before we can consider your manuscript for acceptance. Your revisions should address the specific points made by each reviewer.

[LINK]

Yours sincerely,

Harmit S. Malik

Associate Editor

PLOS Genetics

Kirsten Bomblies

Section Editor: Evolution

PLOS Genetics

Reviewer's Responses to Questions

**Comments to the Authors:**

Reviewer #1: Review uploaded as attachment

Reviewer #2: In this manuscript, Zhao et al. characterize a structural rearrangement in the rcan-1 gene in C. elegans, and link this genetic change to alterations in foraging behavior and fitness. First, they use a panel of N2/LSJ2 RILs to investigate how genetic changes impact fitness, using a competitive fitness assay. They identify a single strain that appears to be an outlier and then show that this strain also displays in increase in exploratory foraging behavior. They then use genetic mapping and sequencing to show that this strain has a de novo genetic change in the rcan-1 gene that is linked to the fitness and foraging phenotypes. Zhao et al. characterize the genetic change in rcan-1 and show that it is a surprisingly complex genomic rearrangement that arose during generation of the RILs. Genetic studies show that the change in rcan-1 is causal of the fitness and foraging phenotypes. Moreover, they find that the rcan-1 genomic rearrangement alters expression levels of rcan-1.

Overall, this is an interesting and convincing paper that should be of broad interest to the readers of Plos Genetics. This detailed study of a new genomic rearrangement and its strong ties to fitness and behavioral changes provides new insights into how structural variants cause phenotypic variability. I suggest just a few ways in which the manuscript can be improved and indicate the degree to which this would be preferred (vs. necessary), in my view.

1. The authors characterize the structural rearrangement in rcan-1 through long read sequencing and confirm that the Illumina sequencing aligns well to the proposed rearrangement. They indicate that they were unable to PCR the entire region. If possible, I’d still suggest a series of smaller PCRs aimed at confirming their proposed rearrangement. I realize this is non-trivial given the repetitive nature of the rearrangement, but perhaps primers could be designed to span the new sequences at breakpoints (i.e. right around the chimeric reads). Overall, they already have strong evidence for their proposed rearrangement, but this structural variant is the central point of the paper and it would be preferable to have overwhelming confidence in their interpretation.

2. The authors provide a solid first pass analysis of how the rearrangement of rcan-1 causes the foraging and fitness phenotypes, by comparing the NILs to a null mutant and examining how the non-coding changes impact expression (using fluorescent reporters). However, this is obviously still not fully resolved. For example, do the two truncated rcan-1 copies have any functional roles? Is this simply a gene dosage effect (seems unlikely…)? There are further analyses that could provide insights: A) what is the phenotype of the heterozygous NIL? B) Do CRISPR induced frameshift mutations in each of the four ORFs that comprise the genomic rearrangement impact the NIL phenotype? If possible, I’d suggest further exploration of this topic, but I do not view it is absolutely necessary for publication (especially given the challenges of targeting these repetitive copies via CRISPR, etc).

Minor points:

1. Is the distribution of datapoints in Fig. 3B bimodal, as would be expected from a single causal variant? It is difficult to assess this as presented. Please provide a histogram or something to that effect.

2. The images in Fig. 6C are rather small and difficult to assess. I’d suggest larger, high-quality images that give the reader a more complete view of sites of expression. In addition, providing a well-labeled image for each of the promoter regions used would be helpful for future mechanistic studies of rcan-1 function.

Reviewer #3: In this manuscript, the authors investigate the genetic basis of increased fitness in a single C. elegans outlier recombinant inbred line (RIL) that has higher fitness than its parental strains. This higher fitness is correlated with increased exploration of a bacterial lawn. The authors provide compelling data that show that the major causative allele contributing to these phenotypes is a de novo complex structural rearrangement of the rcan-1 gene involving a series of tandem inversions/duplications that ultimately leads to reduced expression of rcan-1 due primarily to truncations of the rcan-1 5’ noncoding region. Though the paper does not explore the specific cellular mechanisms by which this rearrangement and rcan-1 affect fitness/exploration, it is a very interesting and clearly-written study based on solid data and is appropriate for PLOS Genetics. In particular, the genetic and long-read sequencing data to map the rearrangement and determine its structure are very strong. We suggest a few experiments that could strengthen the paper, but do not feel that these are critical for publication.

Considerations:

1. One of the interesting conclusions of the paper is that even though the rearrangement creates two full-length copies of the rcan-1 coding sequence, there is actually reduced rcan-1 expression (~25% normal total levels). This is supported by solid RNA-seq data and expression reporters (Fig. 6). Given that the 5’ regulatory regions of both intact rcan-1 ORFs are truncated, reduced expression makes sense. The simplest model is that the truncations remove important enhancers and these enhancers no longer function as well when present more distantly in the remaining intact 5’ regulatory region. However, the authors push the data further to argue that there are tissue-specific effects on expression and that the two orientations of the inverted and truncated 5’region have different effects on expression. These latter two conclusions are not as well supported by the existing data that are based on expression reporters of the truncated/inverted 5’ regulatory region in the two orientations (Fig. 6B-D). The problem is that these expression reporters are extrachromosomal arrays that are overexpressed and are often variable in structure and expression levels from line to line or even different animals of the same line and thus should not be used for the kinds of fine quantitative inferences the authors make, such as the head expression being more affected than the body, especially as these inferences are based on a very small number of lines. To be able to make such fine comparisons the authors should generate single-copy transgenes that are not prone to variability. Additionally, the images in Fig 6C&D do not clearly show the distinct and tissue-specific effects the authors claim, which could again be the result of analysis of a small number of selected lines. Higher magnification images with better quantitation of expression in specific cells of interest (such as the unaffected head neurons) could also help.

We suggest that the authors either repeat these expression experiments with single-copy transgenes or tone-down their conclusions (lines 337-349, especially lines 347-349). For the data in Fig. 6B, they should specify whether the ~30 animals analyzed come from one or more transgenic lines. Three lines of each are shown in the strain list, but it isn’t clear whether the data come from all three lines. In the raw data (Table S1), please list the strain and allele # of each animal analyzed as well.

2. The authors provide good evidence that the residual expression of rcan-1 in the rearrangement is important for increased fitness by showing that near-isogenic lines (NILs) carrying this rearrangement outcompete a rcan-1 null mutant (Fig. 6F). The rcan-1 null has increased exploration (Fig. 6E) compared to a WT control, but it is unclear whether the null mutant also has increased fitness. To test this, a good experiment would be to compete the rcan-1 null vs. the WT control. This experiment would further address the argument that reduced expression of rcan-1 increases fitness and whether fitness is indeed tightly correlated with exploration.

It is clear that some of the complexity of the rearrangement is necessary for its fitness advantage, but it is unclear how much of the complexity is necessary. The simplest model is that the two intact rcan-1 ORFs with their altered upstream regulatory regions would be sufficient – that reduced rcan-1 expression plus some residual expression (possibly tissue-specific) is sufficient. To test this, a strain could be generated with the two altered 5’ regions R1 and R2 driving the rcan-1 ORF (as single-copy transgenes to control expression levels) in the background of the rcan-1 null mutant. Would this strain fully recapitulate the phenotypes of the more complex rearrangement lines?

Other simple experiments could further probe the relationship between rcan-1 dosage and exploration/fitness and strengthen the conclusions. The rearrangement has about 25% normal total expression of rcan-1, though this may be higher in some cells and lower in others. What is the phenotype of a rcan-1(lf)/+ heterozygote that would be predicted to have 50% rcan-1 levels in all cells? What is the phenotype of the rearrangement when heterozygous? Is the rearrangement rescued (i.e. fitness and exploration decreased) by injection of WT rcan-1(+)?

3. The NILs are shown to be sufficient for increased exploration and fitness (Fig. 5). But do they fully account for the increased fitness of the original RILhf strain? To test this, a good experiment would be to compete the NILs with RILhf.

4. On several occasions, it is suggested that variation on chromosome V contributes to the exploration phenotype of RILhf (e.g. lines 190, 261). Though the LSJ2/RILhf RILS clearly show an effect of chromosome V (LSJ2 alleles on this chromosome are associated with increased exploration), it seems that variation on this chromosome can contribute to exploratory behavior, but it seems unlikely that such variation is contributing to the phenotype of RILhf since it has no LSJ2 alleles on chromosome V (perhaps RILhf would explore even more with LSJ2 alleles on chromosome V, but the assay used here is already maxed out so wouldn’t be able to resolve this). More likely, alleles on LSJ2 chromosome V have increased exploration independently of the rest of the RILhf genetic background, a possibility that could be easily tested if desired. However, this is really a fairly minor point and tangential to the paper. We suggest just making it clear in the writing that chromosome V variation may affect exploration, but not necessarily the exploration phenotype of RILhf.

5. Some simple investigations into the modified regulatory sequence would be interesting. Are there known transcription factor binding sequences or conserved sequences in the 5’ noncoding region that are lost in the truncations?

6. A deeper discussion of worm exploration behavior could enrich this manuscript. What is known about the circuitry controlling this behavior? Is rcan-1 expressed in specific relevant neurons (and expression lost in the R1 and R2 constructs)? Additionally, it is ultimately unclear how exploration relates to fitness and whether this correlation is causal or coincidental. Throughout the paper, we are led to believe that the altered exploration could lead to increased fitness by changing how worms feed (the exploration behavior is sometimes called a “foraging strategy” but perhaps more care should be taken with these terms). But late in the discussion (line 405), we learn that there is no difference in food consumption in strains carrying the rcan-1 rearrangement. An interpretation of this surprising result seems warranted. Does this mean that exploration is unrelated to food consumption (and fitness) or might there be other ways that exploration is related to fitness? The authors don’t need to have the answers or take a side, but it would be nice to have some conclusion since it was just confusing to provide this result with no further comment. (Perhaps not calling exploration a foraging strategy would be advisable, especially when increased exploration is considered “increased foraging activity” as in line 363).

7. Is the known functional connection of rcan-1 to tax-6/calcineurin important? If rcan-1 inhibition of calcineurin is relevant to its exploration and fitness phenotypes, then it might be expected that there is increased calcineurin activity in strains with the rearrangement, and a tax-6 mutation would suppress the increased exploration and fitness phenotypes of the rearrangement.

8. There’s a fair bit of discussion of the rcan-1 rearrangement being formed as an adaptation in response to selective pressures (lines 35-38, 368-385, 393-395). Though the rearrangement clearly increases fitness, it does not seem that it was selected for increased fitness, but rather was likely just formed and fixed accidentally. In fact, the way the RILs were made, there did not seem to be any obvious selection for fitness since about half of the lines have low fitness like their parent LSJ2 (Fig 1D). We recommend a more conservative discussion of these points.

Minor points:

1. The authors are to be commended for using box plots rather than bar graphs in the figures (and for providing the raw data in a supplemental file), but the figure legends should give n values for quick reference.

2. It is unclear why the authors measured animal length (Figure 5D). This is the only time we hear of this phenotype in the paper and it is unclear if it is considered to be an important contributor to increased fitness. As presented, the experiment doesn’t really seem to fit in the paper, but is just a random piece of data thrown in.

3. For Fig. 6A and S4, it would be nice to know which other genes have significantly altered expression (the green and red dots), and whether the same genes were affected in both NILs. This information is not easily derived from the raw data in Table S3.

4. The way the Fig 6A legend is written, it is unclear whether this figure shows combined data from the two NILs (“differences between the NIL and N2* strains”). Given that the figure itself is labeled NIL1, we presume that Fig 6A shows the data from NIL1 and Fig S4 shows the data from NIL2. Please make this clear.

5. The Fig 6A legend should say what the green and red dots mean. The Fig 6C&D legend should say what the arrowheads indicate. It would be best if all the images in Figs 6C and 6D are shown with the same anterior-posterior orientation and mention this orientation in the legend. It seems that the zoomed-in images in 6D show different animals than in 6C, but this should be made explicit.

6. In Figure 4A, the schematic for the gene position is difficult to interpret. It would be good to color code rcan-1 differently from pst-2 so it can be seen where these genes start and end, and show the direction of transcription of each gene (as in Fig 4C). In fact, we recommend using the same schematic in 4C and 4A so that they can be directly compared, especially for the 5’ noncoding region. A scale bar for Fig 4A would also help so we can more easily see how much of the 5’ region has increased coverage.

7. In Figure 6B, the significance bars for statistical comparisons are difficult to interpret, especially the top two levels of these comparisons. It is unclear which data sets are indicated as significantly different from each other. For instance, what do the two *** values at the top refer to? The head of WT compared to head of R1 and R2, the body of WT compared to the body of R1 and R2, etc? Or head, body, and total of WT compared as a combined group to head, body, and total of R1 and R2?

8. It would be interesting to see if there are differences in movement speed between the N2*, RILhf , and rcan-1 NILs. This may be linked to exploratory behavior.

9. The description of the structural changes in the rcan-1 rearrangement is useful but could be condensed. The paragraph at the bottom of page 9/top of page 10 repeats many of the same things said in the opening paragraph of page 9 where the rearrangement is first described.

10. There should be explanation for what the different colors mean in Figures S1 and S3.

11. The acronym “CNV” should be defined when first used in the abstract.

12. In Table S3, rcan-1 is written as rcn-1. Please correct this to facilitate searching for the data.

13. Line 93 should read: “in wild strains of C. elegans” instead of “in wild strains C. elegans.”

14. There is inconsistency on whether the rearrangement carries five or six tandem inversions (e.g. lines 25 and 213).

15. Line 366. The rearrangement is described as “similar” to a previously published rearrangement that causes another phenotype in C. elegans. However, it’s not clear what is meant by “similar” and this other rearrangement is in fact quite different. It consists of several duplications and a triplication, but no inversions, and causes a phenotype through increased gene dosage rather than decreased expression as reported here. Additionally, the other rearrangement seems to be formed by chromoanasynthesis in which the rearrangement is made by templated synthesis and the breakpoint junctions have short microhomologies. This seems quite different from the rearrangement described here, which does not seem to have any microhomologies at the breakpoints (though this could be explicitly mentioned). The only similarity seems to be that complex rearrangements can cause phenotypes, but we would argue that the rearrangements themselves are not similar at all.

16. Lines 383-385: the point of this sentence is unclear.

17. The paragraph in lines 407-412 seems overly speculative.

18. References 56 and 58 are the same (and can be updated now that the paper is published).

Reviewed (and signed) by Michael Ailion and Lews Caro

**Have all data underlying the figures and results presented in the manuscript been provided?**

Reviewer #1: Yes

Reviewer #2: Yes

Reviewer #3: Yes

PLOS authors have the option to publish the peer review history of their article (what does this mean?). If published, this will include your full peer review and any attached files.

Reviewer #1: No

Reviewer #2: No

Reviewer #3: Yes: Michael Ailion and Lews Caro

---

## [Decision Letter · Decision Letter 1]

24 Dec 2019

Dear Dr McGrath,

Thank you very much for submitting your Research Article entitled 'A spontaneous complex structural variant in rcan-1 increases exploratory behavior and laboratory fitness of C. elegans' to PLOS Genetics. Your manuscript was fully evaluated at the editorial level and by independent peer reviewers. You will see that all three reviewers are happy with your revisions. However, I hope you will take their comments about the writing in some of the 'new' parts to heart- I agree with Reviewer 3 that some of the changes you have made are not as nicely written as the original pieces. This is your opportunity to smooth those rough edges out before your article is published. I hope you will take that opportunity to do so. Essentially, this is just that- an opportunity to slightly edit otherwise your article is 'accepted' in principle.

We therefore ask you to modify the manuscript according to the review recommendations before we can consider your manuscript for acceptance. Your revisions should address the specific points made by each reviewer.

[LINK]

Yours sincerely,

Harmit S. Malik

Associate Editor

PLOS Genetics

Kirsten Bomblies

Section Editor: Evolution

PLOS Genetics

Reviewer's Responses to Questions

**Comments to the Authors:**

Reviewer #1: The authors have identified an interesting laboratory-derived strain (RILhf) with a spontaneous mutation that causes increased fitness and roaming behavior. My primary concern with the manuscript was that the claim that the rcan-1 rearrangement is causal for the increased fitness and roaming behavior needed to be strengthened. The authors have attempted to address this concern, and the revised Table S3 is excellent and addresses my questions about the backcrossed NILs.

However it is disappointing that rescue of RILhf with wildtype rcan-1 did not work, with respect to foraging behavior (Figure S8). In the author’s “response to reviewers” this experiment is interpreted clearly, but I have some concerns with the authors interpretation in the paper (lines 300-302). My concerns are as follows:

“potentially this indicates that the rearranged region causes ectopic expression necessary for the changes in foraging behavior…” – this is inconsistent with the rcan-1 CRISPR KO allele promoting foraging behavior.

“…or overexpression of rcan-1 decreases foraging behavior” – this is the opposite of what the data is showing, so perhaps a typo? Do the authors mean overexpression of rcan-1 might increase foraging behavior? Otherwise you would have expected the construct to rescue, since RILhf displays increased foraging behavior.

I think more likely is this particular transgenic experiment didn’t work, either for technical reasons, or because additional variations in RILhf promote foraging behavior independent of the rcan-1 rearrangement. Nevertheless, the manuscript deserves to be published in PLoS Genetics, as the rcan-1 rearrangement is a very complex mutation, and fully dissecting the genetics of the various rcan-1 duplications is outside the scope of this paper. Additionally, the findings are interesting and novel, and the experiments are performed carefully and not over-interpreted.

Typo in line 94: “in certain context”

Typo in lines 238-239: should delete “enhancers and other r promoter regions are probably present in the truncated promoters”

Reviewer #2: The authors have addressed all of my concerns.

Reviewer #3: I support publication of this paper, which has been significantly strengthened in this revision. The authors have done a very good job of responding to and addressing the majority of the critiques. Though I do not have major concerns, a few of my original points have not been fully addressed and I repeat them here. Also, many of the parts of the manuscript that were changed upon revision are not written with the same quality as the rest of the manuscript. I point out a number of clear grammar mistakes and typos, but suggest the authors look again at the writing in all their new or edited sections and work more carefully on these parts.

Points:

1. The authors have done a better job of toning down their conclusions about tissue-specific effects of the altered rcan-1 promoter regions in some places and pointing out some of the caveats of extrachromosomal arrays, but there are other places where they still explicitly state that there are tissue-specific effects: Fig 6 title (line 456), line 116, lines 277-279. I am fine with the authors suggesting there might be tissue-specific effects, but think they should not flat out state it as if it were demonstrably proven because the evidence is quite weak.

2. I strongly recommend that the authors not use the term “foraging behavior” for the exploration phenotype and not equate increased exploration with “increased foraging activity.” It isn’t exactly clear what the increased exploration of rcan-1 variants indicates and whether it is causally related to the increased fitness, but it does not seem clearly related to foraging for the rcan-1 variants as their increased exploration does not lead to increased food consumption. Thus, it would be more conservative to just call it “exploration” as that is what is being assayed. The word foraging means “searching for or obtaining food.” The evidence given is that rcan-1 does not affect “obtaining food” and I don’t think it is addressed whether the exploration of rcan-1 variants indicates “searching for food.” An alternative unexplored possibility is that the rcan-1 variants may simply have increased locomotory activity or decreased reversals. Though increased exploration may be a foraging strategy for other mutants, the evidence in this paper does not suggest it is for the rcan-1 variants. Thus, continued use of the term foraging to describe this behavior is both potentially confusing and misleading. Examples include: the short title of the paper, line 51, lines 168-172, line 254, lines 293-308, lines 315-316, lines 349-353, and several Figure legends, but there may be others.

3. Related to point 2, I think the food consumption experiment would fit better in the Results, in the same section with the experiment using uniform bacterial lawns, as a general consideration of the question of how exploration relates to foraging.

4. Though the authors now state that the rcan-1 rearrangement was unlikely to have been selected for its higher fitness, the sentences in lines 38-41 and 346-348 still make it sound that this rearrangement formed as an adaptation in response to selective pressures. I would tone these down.

5. I previously suggested that the paragraph in lines 371-376 is overly speculative and the authors responded by saying that they have toned this paragraph down, yet it reappears exactly the same as in the original.

6. I did not realize that body size is considered a fitness proximal trait, and am surprised that a smaller size would be associated with increased fitness (if anything, I would have thought the opposite). Thus, the experiment on body length could be better explained and motivated for naïve readers like me. For me, it still seems to come out of the blue, and the immediate segue from short body to increased fitness (line 230) is not intuitive at all.

7. In Figure 5D (body length), the y-axis is unclear. It shows “normalized body length” but the units are unclear and it is unclear what it is being normalized to – the legend says “controls” but it is unclear what these controls are and there are no apparent data for any control strains in the Table S1 source data file.

8. Figure 6C&D: please indicate anterior-posterior direction of these worms.

9. The rcan-1 rearrangements suggests a simple and interesting model for the evolution of gene duplicates that wasn’t clearly presented in the discussion of this phenomenon (lines 321-338). How do gene duplicates evolve from redundancy to become paralogs? The rcan-1 rearrangement suggests a model that by imperfectly duplicating or rearranging the regulatory regions of the duplicated genes, two new genes can be created in a single step that aren’t functionally redundant because they may have differential expression levels and differential tissue-specific expression. Thus, this mechanism might create paralogs instantaneously without having to first evolve through redundancy.

Typos/grammar:

line 49: understanding

line 167: causal

line 230: these data (should be plural)

line 237: should be “upstream sequence”

lines 237-242: run-on sentence (also kind of unclear, I have a feeling it may have changed in ways unintended)

line 238: r at end of line

line 281: fluorescence

line 307: solely

lines 317-318: I think the cited paper has just one rearrangement, so shouldn’t be plural

line 319: unclear antecedent of “this.” Just say: “The rcan-1 rearrangement”

line 335: grammar

line 337: RILhf

line 342: “single single-nucleotide” is awkward

line 378: described

line 378: should be “often” instead of “common”

line 466: compared

lines 467-468, 470-471: grammar problems. Also, say what the white arrows in the R2 panels are showing.

lines 460 & 1045: Table S5

lines 1053 & 1076: grammar problems

Table S3 title: says Table S2 by mistake

Figure S4: rearranged

Figure S7: “deletion” is written on top of “region”

Reviewed (and signed) by Michael Ailion.

**Have all data underlying the figures and results presented in the manuscript been provided?**

Reviewer #1: Yes

Reviewer #2: Yes

Reviewer #3: Yes

PLOS authors have the option to publish the peer review history of their article (what does this mean?). If published, this will include your full peer review and any attached files.

Reviewer #1: No

Reviewer #2: No

Reviewer #3: Yes: Michael Ailion

---

## [Editor Report · Decision Letter 2]

11 Jan 2020

Dear Dr McGrath,

We are pleased to inform you that your manuscript entitled "A spontaneous complex structural variant in rcan-1 increases exploratory behavior and laboratory fitness of C. elegans" has been editorially accepted for publication in PLOS Genetics. Congratulations!

Yours sincerely,

Harmit S. Malik

Associate Editor

PLOS Genetics

Kirsten Bomblies

Section Editor: Evolution

PLOS Genetics

Comments from the reviewers (if applicable):

**Data Deposition**

http://datadryad.org/submit?journalID=pgenetics&manu=PGENETICS-D-19-01356R2

**Press Queries**

---

## [Editor Report · Acceptance letter]

12 Feb 2020

PGENETICS-D-19-01356R2 

A spontaneous complex structural variant in rcan-1 increases exploratory behavior and laboratory fitness of C. elegans 

Dear Dr McGrath, 

We are pleased to inform you that your manuscript entitled "A spontaneous complex structural variant in rcan-1 increases exploratory behavior and laboratory fitness of C. elegans" has been formally accepted for publication in PLOS Genetics! Your manuscript is now with our production department and you will be notified of the publication date in due course.

With kind regards,

Kaitlin Butler

PLOS Genetics

On behalf of:
